# MATHPILE : A Billion-Token-Scale Pre-training Corpus for Math

**Zengzhi Wang**[1,3,4]    **Xuefeng Li**[1,4]    **Rui Xia**[3]    **Pengfei Liu**[1,2,4]*

[1]Shanghai Jiao Tong University    [2]Shanghai Artificial Intelligence Laboratory
[3]Nanjing University of Science and Technology    [4]Generative AI Research Lab (GAIR)
zzwang.nlp@gmail.com    pengfei@sjtu.edu.cn

## Abstract

High-quality, large-scale corpora are the cornerstone of building foundation models. In this work, we introduce MATHPILE , a diverse and high-quality math-centric corpus comprising about 9.5 billion tokens. Throughout its creation, we adhered to the principle of "*less is more*", firmly believing in the supremacy of data quality over quantity, even in the pre-training phase. Our meticulous data collection and processing efforts included a complex suite of preprocessing, prefiltering, language identification, cleaning, filtering, and deduplication, ensuring the high quality of our corpus. Furthermore, we performed data contamination detection on downstream benchmark test sets to eliminate duplicates and conducted continual pre-training experiments, booting the performance on common mathematical reasoning benchmarks. We aim for our MATHPILE to boost language models' mathematical reasoning abilities and open-source its different versions and processing scripts to advance the field (available at `https://github.com/GAIR-NLP/MathPile/`).

## 1 Introduction

High-quality, diverse pre-training corpora form the cornerstone for developing powerful foundation models, enabling AI assistants like ChatGPT [47] to exhibit balanced competencies across a broad spectrum of tasks [11]. In this work, our concern centers on mathematical reasoning capabilities within foundational language models [13, 5, *inter alia*], for which can potentially boost the application in education tools, automated problem solving, data analysis, etc., thereby improving user experience. To facilitate this, we are not directly building a model, but rather focusing on a more fundamental aspect: *creating a high-quality and diverse pre-training corpus tailored for the math domain*, namely MATHPILE . Specifically, our work is significantly different from the previous work in the following characteristics (cf. Table 1 for comparison):

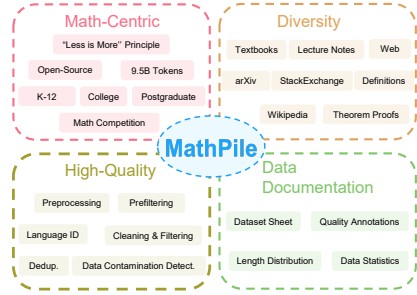

Figure 1: Key features of MATHPILE .

**Math-centric**. Previous open-sourced pre-training corpora have typically focused on general domains, such as Pile [20], RedPajama [59] and Dolma [2]. Others have concentrated on multilingual aspects or programming languages, such as ROOTS [32] and The Stack [30], respectively. However, a notable absence in these offerings is a corpus specificlly tailoring for mathematics. While there exist some corpora designed for training or continually improving math-specific

---

* Corresponding author

38th Conference on Neural Information Processing Systems (NeurIPS 2024) Track on Datasets and Benchmarks.

language models, such as Minerva's mathematical training dataset [35] and OpenAI's MathMix [37], these are not open-sourced. Note that a recent work concurrent with ours, OpenWebMath [49], although math-centric, is solely sourced from web pages. We will discuss the comparison with it later. Recognizing this gap, we aim to democratize access to high-quality mathematical corpus, fostering inclusive advancements in language models' mathematical reasoning.

**Diversity**. While Hendrycks et al. [26] introduced AMPS, a problem set ranging from elementary mathematics to multivariable calculus (K-12 level) for pre-training purposes, it lacks content at the college-level and more challenging competition-level mathematics, focusing instead on a supervised dataset rather than an extensive corpus. The ProofPile corpus, introduced by Azerbayev et al. [4], aims to improve autoformalization and formal proving capabilities in models, yet its scope is confined to formal proving, not covering the broader mathematical domain from K-12 to postgraduate level. Concurrently with our work, Paster et al. [49] propose the OpenWebMath corpus, featuring a corpus composed of mathematical web pages. However, our corpus goes beyond web pages, integrating high-quality mathematics textbooks, lecture notes, scientific papers from arXiv in the field of mathematics, and carefully selected content from StackExchange, ProofWiki, and Wikipedia among others, which positions our corpus as a richer and more diverse mathematical resource for language models.

**High-Quality**. Recent studies have increasingly highlighted the detrimental effects of low-quality and repeated content in pre-training corpora on model training, as evidenced in various works [1, 42, 33, 27, 41]. The importance of high-quality datasets has thus come to the fore. It has been shown that properly filtered and deduplicated web data can yield models as equally powerful as those trained on curated, high-quality corpora [51]. This similar practice has been recently adopted in several notable studies [12, 2, 60]. A notable example is the 1.3 billion-parameter code-focused model pre-trained on synthetically generated textbooks and filtered web pages, a project that broke existing scaling laws although did not open source its data [24]. It's important to emphasize that quality of the corpus is far more significant than its quantity. For instance, OpenAI's MathMix comprises only 1.5 billion tokens. In this work, we diligently adhere to the principle of *less is more*, as outlined in Zhou et al. [71]. To achieve a high-quality corpus, Unlike other approaches that uniformly process all data, we have conducted specialized preprocessing and prefiltering for each data source before global data processing (including language identification, filtering, cleaning, and deduplication). We're dedicated to refining and optimizing our corpus, making a distinctive contribution to the field.

**Data Documentation**. Auditing large-scale pre-training corpora is essential for identifying content characteristics, intended uses, and potential biases, despite challenges due to their size [7, 21, 44]. However, many such corpora are released without proper documentation [45]. Recent audits of certain pre-training corpora have uncovered issues such as irrelevant content [42, 31, 19], copyright infringement [6], and inclusion of test sets for downstream tasks [1, 18], highlighting the need for detailed data sheets and transparent documentation. To this end, following previous efforts to enhance corpora transparency, we have provided a dataset sheet for our MATHPILE (see Table 6). Throughout our extensive data processing workflow, numerous documents were annotated for quality, such as language identification scores and the ratio of symbols to words (as exemplified in Figure 12). These quality annotations enable future users to apply their specific filters based on these scores. Additionally, we have conducted extensive deduplication for this corpus and performed data contamination detection with downstream benchmark test sets, removing any duplicated samples identified (cf. § 3.4). Interestingly, we have also discovered a significant number of questions from downstream test sets in OpenWebMath (cf. § 3.4). This underscores the importance of meticulous data documentation. We plan to release different versions to facilitate future use. See Appendix C for examples.

Additionally, we conducted continual pre-training experiments on MATHPILE and found that it generally enhances the performance of language models across various mathematical reasoning benchmarks, with an average improvement of up to 5% (cf. § 4.2). In conclusion, we hope to facilitate the growth of the field of AI for mathematics by contributing this specialized, high-quality, diverse corpus focused on the mathematical domain while maintaining utmost transparency about the data for practitioners.

## 2 The Collection of Corpora

In order to construct MATHPILE , we gather data from a variety of sources, which also includes a component of manual collection. We provide an ethics statement regarding copyright in Appendix B.

Table 1: The comparison of MATHPILE with other mathematical Corpora, where PS denotes the problem set type. For non-open-sourced corpora, details are inferred from literature, with unknowns marked as "?". Token counts may vary by tokenizer; we use statistics from each dataset's report and the GPTNeoX-20B tokenizer [9] for our corpus. DM-Mathematics is from Saxton et al. [56]. "Minerva" refers to its dataset. ProofPile-2 [5], encompassing OpenWebMath and others, is excluded from this comparison.

| Datasets | Open Source | Type | Target Domain | # Textbooks | Has Synth. Data | Data Contam. Detection | # Tokens | Source |
|---|---|---|---|---|---|---|---|---|
| Minerva | ✗ | Corpus | General Math | ✗ | ✗ | ✓ | 38.5B | arXiv, Web |
| MathMix | ✗ | Corpus + PS | General Math | ? | ✓ | ✓ | 1.5B | ? |
| ProofPile | ✓ | Corpus | Theorem Proving | 7 | ✗ | ✗ | 8.3B | arXiv, Textbooks, Lib., StackExchange, ProofWiki, MATH |
| OpenWebMath | ✓ | Corpus | General Math | ✗ | ✗ | ✗ | 14.7B | Web |
| DM-Mathematics | ✓ | PS | Math Competition | ✗ | ✓ | - | 4.4B | Synthesis |
| AMPS | ✓ | PS | Math Competition | ✗ | ✓ | ✗ | 0.7B | Khan Academy, Synthesis |
| MATHPILE (Ours) | ✓ | Corpus | General Math | 3,979 | ✓ | ✓ | 9.5B | arXiv, Textbooks, StackExchange, Wikipedia, ProofWiki, Web |

**Mathematical Textbooks**  Textbooks, covering mathematical concepts, exercises, and solutions, are valuable for *educational purposes* for both humans and machines. Recent studies, even though not focused on math, support this view with synthesized textbooks [24, 36]. To collect these genuine and high-quality textbooks, we began by conducting extensive manual searches across the internet, seeking open-source and freely accessible mathematics-related textbook websites. Afterwards, we proceeded to download these PDF files, resulting in a collection of 38 K-12 level textbooks, along with 369 college-level mathematics textbooks that cover a wide range of subjects including linear algebra, probability theory, calculus, and optimization. In addition to these textbooks, we also included 467 college course handouts and lecture notes, which tend to be more concise compared to full-length textbooks. Subsequently, we employed the Mathpix API[2] to parse the PDFs into markdown format. Then, we meticulously cleaned up extraneous elements such as parsed image URLs, preface sections, table of contents, acknowledge sections, index sections, and consecutive empty lines within the parsed content, resulting in a total of 874 documents.

We also refined high-quality mathematics-related synthetic textbooks from OpenPhi Project.[3] It is an open-source counterpart to the Phi work [24]. While the underlying model and generation process differ, the output encompasses a broad spectrum of subjects, extending beyond programming. To isolate mathematics-related documents, we employed a straightforward criterion: the presence of the symbol "$$" combined with common mathematical expressions like "\{mathbf" and "\{frac". While "$$" alone is not always reliable, combining it with these symbols improves accuracy based on manual verification. This approach yielded 3,889 documents from an initial pool of 124,493. As the volume of pre-training data escalates, the synthesis of high-quality data becomes increasingly crucial. More advanced filtering methods and mathematical corpora synthesis are left for future exploration.

**Mathematical Papers from ArXiv**  ArXiv offers a free distribution service and serves as an open-source archive housing millions of scientific papers. It also provides invaluable training data for numerous powerful language models [61, 59, *inter alia*]. In our endeavor to collect mathematical papers from ArXiv, we identify 50 sub-subjects spanning Mathematics, Computer Science, Statistics, Physics, Quantitative Finance and Economics. Our process involved filtering ArXiv's metadata[4] to focus on the chosen subjects (cf. Table 7), followed by accessing the source LaTex files (if available). We exclusively retained the LaTex files and consolidated multiple files based on their respective order as indicated by commands such as "include" and "input" within the main LaTex file of each paper. Subsequently, we undertook extensive transformations to enhance data clarity and consistency, including removing comments, reverting macros, omitting figures but keeping captions, excluding acknowledgements and references, condensing empty lines, replacing some formatting commands, substituting titles, and preserving only the main body content (cf. § D for more details). Finally, we compiled 347,945 meticulously cleaned LaTex documents (around 8.5 billion tokens), with each document corresponding to a single paper.

---

[2]https://mathpix.com/ocr

[3]https://huggingface.co/open-phi

[4]https://www.kaggle.com/datasets/Cornell-University/arxiv

**Mathematical Entries in Wikipedia**    Wikipedia is one the largest and most popular free online encyclopedias, offering information on a wide range of topics, including history, science, technology, culture, and more. This extensive knowledge has proven to be highly beneficial for numerous natural language processing tasks [34, *inter alia*] and pre-trained language models [17, 61, *inter alia*]. To collect mathematical entries from Wikipedia, we downloaded the mathematics-focused (without pictures) dump of Wikipedia in English for the month of August 2023. We extracted the HTML documents from the dump using the library `libzim`, resulting in approximately 106,900 documents. Subsequently, we converted these HTML documents into markdown format using the `html2text` library[5] while removing the hyperlinks following the practice of LLaMA [61]. We retained the alternative text content but excluded image (often in SVG format) paths. Additionally, we eliminated extra newlines within paragraphs and condensed more than three consecutive empty lines to two using regular expressions. Further refinement involved the removal of boilerplate content at the bottom of the pages, typically denoted with phrases like "This article is issued from Wikipedia. The text is ...". In the end, our efforts yielded a collection of 106,881 mathematical Wikipedia entries, about 0.8 billion tokens.

**Entries from ProofWiki**    ProofWiki, an online compendium of mathematical proofs, has been instrumental in advancing the fields of autoformalization and formal proof proving, as evidenced by NaturalProofs [65] and ProofPile. We sourced data from the ProofWiki dump dated April 9, 2022 (provided by the Internet Archive), mirroring the preprocessing approach employed by NaturalProofs, which was based on the version from November 12, 2020. Specifically, this involved leveraging the `BeautifulSoup` library to parse all wiki pages followed by the extraction of raw text content using the `wikitextparser` library. This process yielded a substantial collection of mathematical content, totaling about 7.6 million tokens, comprising 10,328 definitions and 13,511 theorem-proof pairs. To facilitate better data organization, we formatted the definitions using the "`definition`" environment, and the theorem-proof pairs within the "`section`" environment with their respective titles serving as the section headings, similar to ProofPile.

**Mathematical Discussions on StackExchange**    StackExchange, renowned for its network of community-powered question-and-answering websites, spans a wide array of topics, each concentrated on a particular topic. Its high-quality data trove has significantly contributed to the development of various language models [61, 71, *inter alia*]. In our study, we identify eleven sites within this network, including five dedicated to mathematics (such as Mathematics and MathOverflow) and six others in closely related fields like Physics (cf. Table 8). Our data collection process began with downloading the site dumps from August 2023 (provided by the Internet Archive). We only retained the essential components in the posts, namely questions and answers (also associated meta information). To convert HTML documents to raw text, we utilized the `BeautifulSoup` library, coupled with a meticulous removal of invalid XML characters. We then systematically paired questions and their respective answers. Each question typically garners multiple responses, each with its own score and in some cases, an endorsement as the accepted answer by the questioner. To guarantee quality, we applied a quality threshold (i.e., 5) for filtering. Questions underwent filtering based on the threshold, whereas answers were assessed by either the threshold or the score of the accepted answer, whichever was lower. Unanswered questions scoring at least 10 were preserved for potential future use. This rigorous process resulted in a rich collection of data, comprising 267,919 questions, 435,129 answers, and 3,418 unanswered questions, totaling about 254 million tokens.

**Mathematical Web Pages from Common Crawl**    Common Crawl, an archive of web data since 2007, is crucial for training advanced language models like GPT-3 [10] and LLaMA. Our work targets extracting math web pages from SlimPajama [12], a cleaned and deduplicated version of RedPajama, focusing on its CommonCrawl and C4 subsets. Eschewing the common approach of using neutral network-based filtering, we opt for heuristic rule-based methods. Our procedure began with the creation of TF-IDF features, derived from our curated high-quality textbooks. During this process, we removed the stop words, limited the features to a maximum of 10,000, and employed white space tokenization. Upon the observation of the resulting vocabulary, we identified 11 commonly used LaTex commands, integral to mathematical expressions. We utilize these commands as a basis for a hard match within each document. A document is classified as mathematical if it contains any of these commands along with the symbol "$$", typically indicative of a mathematical document. This

---

[5]We later found that the `html2text` library resulted in the LaTeX display issue in the cleaned documents (cf. Figure 11). Switching to another library `Resiliparse` with DOM parsing resolved this issue, ensuring correct LaTeX display.

rule-based approach, though simplistic, proved to be highly effective, especially given the vast size of the Common Crawl corpus . We also experimented with more intricate dense embedding-based methods to identify mathematical documents, but these resulted in poor recall. Our efforts resulted in the compilation of a substantial collection of mathematical web pages: 4,307 documents from SlimPajama-C4 and 72,137 documents from SlimPajama-CommonCrawl, totaling approximately 633 million tokens. We acknowledge the potential for more efficient methods to sift mathematical documents from Common Crawl snapshots, an area we plan to explore in future work.

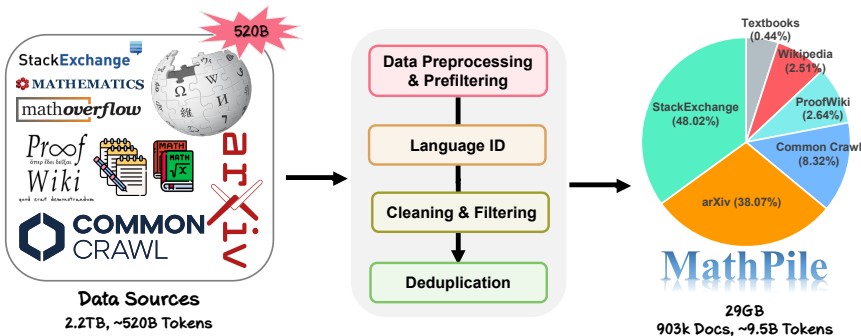

Figure 2: The creation process of MATHPILE . We additionally perform data contamination detection on benchmark test sets (cf. § 3.4). We visualize its component ratios by document counts (Right).

## 3 Global Data Processing

After conducting specific data preprocessing for each data source during the data collection process, we globally engage in three critical steps: language identification, filtering, and deduplication, to ensure the quality of the entire corpus, as shown in Figure 2.

### 3.1 Language Identification

To filter non-English documents, we utilized the fastText language identifier, which was trained on Wikipedia, Tatoeba, and SETimes [29, 23]. A common practice is to classify a document as its respective language if the score exceeds 0.5, a threshold also employed by CCNet [66]. However, during the application of this practice, we encountered a considerable number of false positives—cases where documents were erroneously filtered as non-English when, in fact, they were written in English but contained a substantial amount of mathematical symbols. We attribute this issue to the domain gap between the fastText training datasets and the mathematical content. To enhance non-English document filtering, we set customized score thresholds for each data source. Specifically, Wikipedia and StackExchange thresholds were set at 0.1, arXiv at 0.3, and Common Crawl at 0.5. No thresholds were applied to ProofWiki and Textbooks due to manual verification ensuring English content. This refinement removed about 8,400 documents, totaling 231 million tokens.

### 3.2 Data Cleaning and Filtering

Despite thorough preprocessing, some documents, especially from sources like Wikipedia and Common Crawl, lack quality for language modeling due to brevity or automated content. Existing filtering methods [55, 54, 41, 51, 12], while detailed, risk excluding valuable documents in our math-focused corpus if directly applying them as-is. To address this issue, we developed a unique set of cleaning and filtering heuristic rules, specifically crafted for the mathematical domain and drawing from past studies. These rules are aimed at removing meaningless lines (such as boilerplate content) and documents. Specifically, we (1) detect lines containing "lorem ipsum" and filter them out if the resulting line is less than 5 characters; (2) detect lines containing "javascript" that also include "enable", "disable" or "browser" and are under 200 characters, and filter them; (3) filter lines containing fewer than 10 words that include keywords like "Log in", "sign-in", "read more...", or "items in cart."; (4) filter documents if the ratio of uppercase words exceeds 40%; (5) filter lines that end with "..." if they constitute more than 30% of the entire document; (6) filter documents if the

ratio of non-alphabetic words surpasses 80%; (7) exclude documents with an average English word length outside the range of (3, 10); (8) discard documents that lack at least two common stop words such as "the", "be" "to" "of" "and" "that" or "have"; (9) filter out documents if the ratio of ellipses (...) to words exceeds 0.5 (e.g., progress bars); (10) remove documents where 90% of lines start with bullet points; (11) filter documents including less than 200 characters after removing spaces and punctuation marks.

These meticulously crafted rules enabled us to curate a high-quality mathematical corpus. They also facilitated the assignment of quality annotations to each document from Wikipedia and Common Crawl. These annotations provide researchers and developers with the flexibility to filter the data according to their criteria, catering to specific needs (as shown in Figure 12). This process resulted in filtering approximately 1,100 documents, removing 17 million tokens.

### 3.3 Data Deduplication

Given that our corpus originates from diverse sources, it is inevitable that there will be repetitions both within and across these sources. Deduplication is vital for training efficiency and reducing data memorization, addressing both exact and near-duplicates [33]. We utilized the MinHash LSH algorithm [22] built on the implementation of `text-dup` [46] and Lee et al. [33], to process large-scale corpora efficiently. Specifically, our process involved splitting each document using whitespace and constructing 5-grams, applying the "`sha1`" hash function, and configuring 450 buckets with 20 minhashes each, totaling 9,000 minhashes per document, as per RefinedWeb's guidelines [51].

During the deduplication process within each source, we encountered numerous exact and near-duplicate documents across various sources: 304 in arXiv, 623 in Common Crawl, 83,716 in Wikipedia, 783 in textbooks (primarily synthetic), and 144 duplicate questions in StackExchange. Despite finding many near-duplicates in ProofWiki, they were differentiated as unique lemmas, proofs, or definitions, leading us to retain these entries (cf. Table 13). Manual review revealed significant duplication in Wikipedia due to collecting multiple historical document versions and in StackExchange from reposts across different forums (e.g., Math and MathOverflow) for broader visibility (cf. Table 16). We provide near-duplicate examples from each data source in Table 11-16. Cross-source deduplication revealed minimal overlap, with a single StackExchange question duplicated in Common Crawl, which was removed. This eliminated around 714 million tokens.

Note that we also experimented with using suffix arrays [43] to eliminate exact match sequences within documents. However, it tended to remove common phrases like "Questions: ". While it can effectively remove some templated content, it also disrupts the contextual integrity of our corpus. Consequently, we decided against employing this in order to preserve the context of our data.

### 3.4 Data Contamination Detection

As pre-training corpora grow, encountering data contamination becomes inevitable, where evaluation examples are found in the training set. Traditionally, post-hoc analysis, employing n-gram overlap, assesses contamination levels (e.g., GPT-2 [53], GPT-3 [10], FLAN [63], LLaMA-2 [62]). We advocate for early contamination detection during dataset creation to prevent irreversible damage as delaying exacerbates issues (c.f., previous study [30]). Here, we utilize popular mathematical reasoning benchmarks, namely GSM8K [16], MATH [26], MMLU-STEM [25], AGIEval-SAT-MATH [70], MathQA [3] and AQuA [39] to detect data contamination.

To detect data contamination, we aggregated questions and answers from benchmark tests into a reference set, considering only questions for MMLU, AGIEval, MathQA and AQuA due to its multiple-choice format. Intuitively, math problem solutions often involve diverse reasoning steps, making questions easier to detect for contamination in pre-training data due to their more fixed nature. We utilized line-level exact match detection, dividing documents into lines, hashing each with `MD5` (taking the first 64 bits and the line itself to form sets), and applied this to both our corpus and the test sets. If a test set line and its hash match exactly with our dataset, it's marked as contamination.

Table 2: Benchmark test set occurrences in pre-training corpora, with numbers representing minimum occurrences, given potential undetected duplicates.

| Corpus | MATH | MMLU-STEM |
|---|---|---|
| Ours | 23 | 2 |
| OpenWebMath | 195 | 65 |

After our detection process, we found 23 questions from MATH and 2 from MMLU-STEM in our corpus (see Table 2), with no accompanying answers. No contamination was detected in other benchmarks. These duplicates mainly originated from StackExchange, Textbooks, and Common Crawl (see Table 17 and Table 18 for examples). Notably, questions from AMC mathematics competition books, also used in the MATH benchmark, were identified in Textbooks. We extended our analysis to OpenWebMath, uncovering more duplicate questions from MATH and MMLU (cf. Table 19), although many were repeats. This aligns with similar findings by Azerbayev et al. [5]. These instances highlight the importance of vigilance in creating pre-training corpora to avoid undermining downstream benchmarks. We removed all detected exact matches to mitigate data contamination, resulting in MATHPILE corpus.

## 4 Data Analysis

### 4.1 Statistics

Table 3: The components and data statistics of MATHPILE .

| Components | Size (MB) | # Documents | # Tokens | max(# Tokens) | min (# Tokens) | ave (# Tokens) |
|---|---|---|---|---|---|---|
| Textbooks | 644 | 3,979 | 187,194,060 | 1,634,015 | 256 | 47,046 |
| Wikipedia | 274 | 22,795 | 59,990,005 | 109,282 | 56 | 2,632 |
| ProofWiki | 23 | 23,839 | 7,608,526 | 6,762 | 25 | 319 |
| CommonCrawl | 2,560 | 75,142 | 615,371,126 | 367,558 | 57 | 8,189 |
| StackExchange | 1,331 | 433,751 | 253,021,062 | 125,475 | 28 | 583 |
| arXiv | 24,576 | 343,830 | 8,324,324,917 | 4,156,454 | 20 | 24,211 |
| Total | 29,408 | 903,336 | 9,447,509,696 | - | - | 10,458 |

We present detailed statistical information for each component of MATHPILE in Table 3, such as the number of documents and the count of tokens. Following our meticulous and comprehensive data collection and processing process, we obtain 29GB of high-quality and diverse math-centric corpus, encompassing around 9.5 billion tokens, from an initial volume of 2.2TB of raw data (cf. Figure 2). Compositionally, arXiv constitutes the largest portion of MATHPILE , while Textbooks represent the smallest share but are of exceptionally high quality.

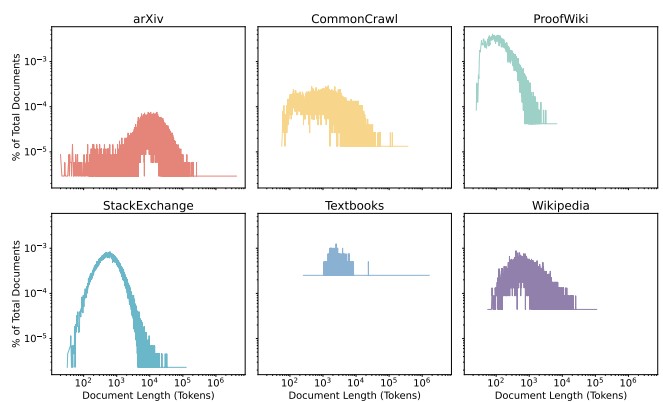

Figure 3: Document length distribution (log-scale).

We analyze the document length (in terms of token numbers) and their respective proportions from each source within MATHPILE , which is visualized in Figure 3. Intuitively, if the data from each source contains a higher amount of near-duplicates or machine-generated content, the distribution of documents of similar lengths becomes more prevalent, leading to a less smooth distribution curve. Figure 3 shows that, thanks to our thorough and rigorous processing, the document length distribution in MATHPILE is relatively smooth across different sources. Note that ProofWiki, due to its fixed format of definitions, lemmas, and proofs, naturally contains shorter content, resulting in a distribution with many similar lengths. We can also observe that, on average, the documents from arXiv and Textbooks tend to be lengthier, while those from ProofWiki and StackExchange are generally shorter.

### 4.2 Continual Pre-training Experiments

We chose `Mistral-7B-v0.1` [28] (the state-of-the-art open-source model at the time) for continual pre-training. We segmented packed text into chunks with a window size of 4,096 and continued

pre-training for 3 epochs with a global batch size of 1024. We employ a cosine learning rate schedule with a maximum learning rate of 1e-5 and 1% warmup steps. All experiments were conducted on NVIDIA A100 8*80GB GPUs. For evaluation, we employ a range of benchmarks - GSM8K, MATH, MMLU-MATH, AGIEval-SAT-MATH, MathQA, AQuA - to assess varying levels of mathematical reasoning abilities, comparing all models using the same few-shot prompting with greedy decoding.

**The Effectiveness of MATH-PILE**  We further pre-trained `Mistral-7B-v0.1` on several subsets, respectively. As shown in Table 4, overall, continual pre-training on the subsets generally enhances performance across diverse math benchmarks, albeit to varying degrees. There are exceptions, such as the lack of improvement on GSM8K after training on StackExchange; we suspect

Table 4: Results on each subset of MATHPILE and sampled OpenWeb-Math. The numbers in parentheses represent the number of tokens trained. **Bold** results denote improvements over the original Mistral.

| Models | GSM8K | MATH | SAT-MATH | MMLU-Math | MathQA | AQuA |
|---|---|---|---|---|---|---|
| Mistral-7B-v0.1 | 47.38 | 10.08 | 47.27 | 44.92 | 23.51 | 27.95 |
| + Textbooks (0.56B) | **48.97** | **12.10** | **56.36** | **48.93** | **30.38** | **33.07** |
| + Wikipedia (0.18B) | **49.96** | 9.96 | **53.63** | **47.16** | **28.97** | **35.43** |
| + StackExchange (0.87B) | 43.06 | **11.66** | 47.27 | 43.51 | **27.67** | **30.70** |
| + Common Crawl (1.83B) | 45.56 | 9.88 | **50.45** | **45.17** | **25.79** | **31.88** |
| + arXiv (0.38B) | **47.91** | 7.50 | 42.72 | **46.34** | 18.05 | 27.55 |
| + Textbooks, Wikipeida, StackEx., CC (4B) | **49.88** | **11.70** | 43.18 | 43.75 | 23.24 | 25.19 |
| + AMPS (1B) | 0.08 | 0.82 | 3.18 | 0.47 | 10.99 | 8.27 |
| + DM-Mathematics (5B) | 0.00 | 0.00 | 0.00 | 0.00 | 0.00 | 0.00 |
| + Sampled OpenWebMath (0.59B) | 43.21 | 7.86 | **47.72** | **47.52** | 21.80 | 24.80 |

this is due to community users rarely asking basic arithmetic questions on StackExchange. Continual pre-training on arXiv leds to a slight performance boost on GSM8K and MMLU-MATH, but a degradation on MATH, SAT-MATH, and MathQA. We attribute this performance degradation to the disparity between the math knowledge present in arXiv papers and that required for the downstream benchmarks. We also conducted pre-training on a collection of Textbooks, Wikipedia, StackExchange, and CC. Experimental results indicate improved performance on GSM8K and MATH, but not on other benchmarks. Due to limited computational resources,[6] we did not extensively experiment with the entire dataset or combine data from MATHPILE 's subsets and existing general corpora, leaving these valuable aspects for future work. Note that we also report some evaluation results on general language benchmarks provided in Appendix G.

Furthermore, we also conducted continual pre-training on some existing corpora listed in Table 1 for comparison, including AMPS, DM-Mathematics and a random subset of OpenWebMath, cleansed of data leakage, in volumes approximately equal to that of Textbooks. Surprisingly, pre-training directly with these synthetic datasets degraded model performance. We attribute this to the narrow, monotonous structure of AMPS and DM-Mathematics problem sets, making them unsuitable for standalone pre-training; such datasets generally yield better results when combined with broader corpora for pre-training [67]. Additionally, the OpenWebMath subset produced even less improvement than the same or smaller scale subsets of MATHPILE , such as Textbooks and Wikipedia (cf. Table 4), likely due to a need for more tokens to show substantial gains. These results underscore the superior quality of our data.

Table 5: Ablation study on data processing pipeline and LaTeX display issue resolution

| Models | Global Data Processing | Fix Latex Display Issue | GSM8K | MATH | SAT-MATH | MMLU-MATH | MathQA | AQuA |
|---|---|---|---|---|---|---|---|---|
| Mistral-v0.1-7B | - | - | 47.38 | 10.08 | 47.27 | 44.92 | 23.51 | 27.95 |
| + Sampled raw Wikipedia (0.55B) | ✗ | ✗ | 41.92 | 6.28 | 20.90 | 23.70 | 24.72 | 24.01 |
| + Full raw Wikipedia (2.18B) | ✗ | ✗ | 32.30 | 4.48 | 13.64 | 25.59 | 27.04 | 23.62 |
| + Full cleaned but LaTeX issued Wikipedia (0.23B) | ✓ | ✗ | 47.15 | 8.58 | 46.81 | 42.92 | 21.00 | 31.88 |
| + Full cleaned Wikipedia (0.18B) | ✓ | ✓ | **49.96** | 9.96 | **53.63** | **47.16** | **28.97** | **35.43** |

**The Effectiveness of Data Processing Pipeline**  We utilized the Wikipedia subset as a testbed to evaluate our data processing pipeline. We distinguished between raw Wikipedia, which is collected but not globally processed, and cleaned Wikipedia, which has undergone global data processing. Additionally, we performed an ablation study on LaTeX display issues in Wikipedia (cf. Figure 11), attributed to HTML-to-text conversion tools, by comparing documents with problematic and correct LaTeX displays. Following previous settings, we executed continual pre-training on these datasets.

---

[6]Pre-training 10 billion tokens for 1 epoch requires approximately 1,760 NVIDIA A100 GPU hours, making us keen to partner with well-resourced corporations to gain deeper insights in the future.

Results in Table 5 indicate that skipping our pipeline notably reduces Mistral's mathematical reasoning abilities, unaffected by increased training size (i.e., 2.18B). Furthermore, correct LaTeX display in documents is vital for enhancing reasoning capabilities, as shown by the last two rows of Table 5. These findings underscore our pipeline's effectiveness and shed light on the superior importance of data quality over quantity, even in the continual pre-training phase.

# 5 Related Work

**Pre-training Corpora for Language Models**   In language modeling, early models like GPT [52] and BERT [17] are trained on resources such as Books [73] and Wikipedia. Later models like GPT-2 [53] and T5 [55] expand training to include web data from Reddit (WebText) and Common Crawl (C4). GPT-3 [10] enlarges its corpus to 300 billion tokens, combining Common Crawl, WebText, Books, and Wikipedia. Pile [20] introduces a diverse collection of 22 datasets for large-scale pre-training. The Gopher project [54] compiles a 10.5TB corpus, and PaLM [14] is built from a 780 billion-token corpus, both closed-source. BLOOM [57] uses the ROOTS dataset [32] for multilingual pre-training. The Stack Kocetkov et al. [30] provides a 3.1 TB code dataset. LLaMA [61] utilizes various data sources but doesn't release its corpus, unlike RedPajama [59] and its de-duplicated version SlimPajama [12]. RefinedWeb shows web-only corpora can rival curated ones [51]. Recent models like GPT-4 [48], Mistral-7B [28] and the lastest Gemini [58] have refrained from open-sourcing data. Constructing diverse, high-quality pre-training corpora is crucial for narrowing the performance gap with closed-source models, reflecting our work's aim.

**Pre-training Benchmarks and Corpora for Mathematical Reasoning**   The challenge of endowing models with human-like mathematical reasoning has attracted significant interest from the machine learning and natural language processing communities. To evaluate models' mathematical capabilities, several benchmark datasets have been developed, including AQuA [38], DM-Mathematics [56], SVAMP [50], GSM8K [16], and MATH [26], which cover a range of complexities from basic arithmetic to competition-level mathematics. Additionally, benchmarks like NaturalProofs [65] focus on theorem-proving capabilities, while the STEM subset of MMLU [25] evaluates understanding across multiple tasks in science, technology, engineering, and mathematics. To improve models' mathematical reasoning, pre-training corpora like AMPS [26] (despite a large-scale synthetic exercise set), ProofPile [4], and OpenWebMath [49] have been introduced, targeting various levels of mathematical problem-solving and theorem proving. Unlike Google's Minerva [35] and OpenAI's MathMix [37], which are not public, our work focuses on creating a high-quality and diverse mathematical corpus from diverse sources to fill existing gaps.

# 6 Conclusion and Limitations

In this work, we present MATHPILE , a specialized corpus centered around mathematics, characterized by its diversity and high quality. Throughout its development, we meticulously source and gather data, applying a rigorous and math-specific pipeline. This pipeline encompasses various stages such as preprocessing, prefiltering, language identification, cleaning and filtering, and deduplication, all aimed at maintaining the high quality of the corpus. We also conduct data contamination detection to remove duplicates from popular mathematical reasoning benchmark test sets, crucial for ensuring their integrity and effectiveness, an aspect often overlooked in other similar works. We aim for our MATHPILE to enhance mathematical reasoning in language models, whether used alone or in conjunction with other datasets, to promote broader applications.

This dataset also has some limitations. Many detailed decisions in its creation were made empirically, which may not always be optimal, and verifying decisions directly can be challenging. Moreover, the data scale is insufficient for training extra-large models; subsets like the common crawl could be expanded. Furthermore, the dataset is focused primarily on English, highlighting the need to construct high-quality datasets for other languages. Future research could also explore data mixing [40] and model-based pre-training corpus refinement [68, 72] to enhance dataset quality and model performance.

## Acknowledgments and Disclosure of Funding

This work was partially funded by the National Natural Science Foundation of China (62476168), Shanghai Artificial Intelligence Laboratory.

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

# Appendix

# A  MATHPILE Datasheet

| MOTIVATION | |
|---|---|
| **For what purpose was the dataset created?** | Developed in a context where datasets like Google's Minerva and OpenAI's MathMix are not open-sourced, MATHPILE aims to counter this trend by enriching the open-source community and enhancing mathematical language modeling with its (relatively) large-scale, math-centric, diverse, high-quality dataset. It can be used on its own or cooperated with general domain corpora like books, and Github code, to improve the reasoning abilities of language models. |
| **Who created the dataset and on behalf of which entity?** | MATHPILE was created by the authors of this work. |
| **Who funded the creation of the dataset?** | The creation of MATHPILE was funded by GAIR Lab, SJTU. |
| **Any other comment?** | None. |

| COMPOSITION | |
|---|---|
| **What do the instances that comprise the dataset represent?** | MATHPILE is comprised of text-only documents, encompassing a broad range of sources. These include academic papers from arXiv, educational materials such as textbooks and lecture notes, definitions, theorems and their proofs, informative articles from Wikipedia, interactive Q&A content from StackExchange community users, and webpages sourced from Common Crawl. All these instances are math-focused. |
| **How many instances are there in total?** | MATHPILE contains about 903 thousand of documents, or around 9.5 billion tokens. |
| **Does the dataset contain all possible instances or is it a sample (not necessarily random) of instances from a larger set?** | MATHPILE is curated from a diverse array of sources, including arXiv, Textbooks, Wikipedia, StackExchange, ProofWiki, and Common Crawl. However, it doesn't encompass all instances from these sources. We have implemented a rigorous data processing pipeline, which involves steps like preprocessing, prefiltering, language identification, cleaning, filtering, and deduplication. This meticulous approach is taken to guarantee the high quality of the content within MATHPILE . |
| **What data does each instance consist of?** | Each instance in MATHPILE is a text-only document, uniquely identified by its source, labeled under Subset. These instances are enriched with metadata, such as the score from language identification, the ratio of symbols to words, and their respective file paths. Note that instances from the StackExchange are composed of a question and its accompanying answers, each with their own set of meta data, including community users. To illustrate them, we provide specific examples for each source, ranging from Figure 4 to Figure 10. |
| **Is there a label or target associated with each instance?** | No. |
| **Is any information missing from individual instances?** | No. |

| | |
|---|---|
| **Are relationships between individual instances made explicit?** | No. |
| **Are there recommended data splits?** | No. |
| **Are there any errors, sources of noise, or redundancies in the dataset?** | Despite our rigorous efforts in cleaning, filtering out low-quality content, and deduplicating documents, it's important to acknowledge that a small fraction of documents in MATHPILE might still fall short of our quality standards, particularly those sourced from web pages. |
| **Is the dataset self-contained, or does it link to or otherwise rely on external resources?** | Yes, MATHPILE is self-contained. |
| **Does the dataset contain data that might be considered confidential?** | No. |
| **Does the dataset contain data that, if viewed directly, might be offensive, insulting, threatening, or might otherwise cause anxiety?** | We do not expect offensive content despite our significant efforts in cleaning and filtering. But, we can not fully guarantee this. |

| COLLECTION | |
|---|---|
| **How was the data associated with each instance acquired?** | Our data is primarily sourced from the arXiv website and the Internet Archive. The CommonCrawl data originates from SlimPajama. The textbooks included are manually collected, with quality checks performed on publicly available textbooks from various internet sources. |
| **What mechanisms or procedures were used to collect the data?** | Refer to § 2 for details on how they collect data. |
| **If the dataset is a sample from a larger set, what was the sampling strategy?** | We strive to use the most recent data dumps available and then selectively choose high-quality documents that are closely related to mathematics. |
| **Who was involved in the data collection process and how were they compensated?** | Authors from this paper were involved in collecting it and processing it. |
| **Over what timeframe was the data collected?** | MATHPILE encompasses documents created between 2007 and August 2023. Note that some documents and textbooks included may be created in the previous century. |
| **Were any ethical review processes conducted?** | No. |

| PREPROCESSING | |
|---|---|

| | |
|---|---|
| **Was any preprocessing/-cleaning/labeling of the data done?** | Yes, during our data collection phase, we conducted extensive filtering and cleansing procedures, detailed in § 2. After the completion of data collection, we conducted further steps including language identification, additional cleaning and filtering, deduplication, and leakage detection in benchmark datasets. Subsequently, we removed any contaminated examples identified through this process. See § 3 for details. |
| **Was the "raw" data saved in addition to the preprocessed/-cleaned/labeled data?** | Yes. |
| **Is the software that was used to preprocess/clean/label the data available?** | Yes, scripts are open-sourced at `https://github.com/GAIR-NLP/MathPile/tree/main/src` |

### USES

| | |
|---|---|
| **Has the dataset been used for any tasks already?** | Yes, this data has been used to develop mathematical language models. |
| **Is there a repository that links to any or all papers or systems that use the dataset?** | No. This dataset is currently utilized in the following research papers: (1) JiuZhang 3.0: Efficiently Improving Mathematical Reasoning by Training Small Data Synthesis Models. (2) Task Oriented In-Domain Data Augmentation. (3) Great Memory, Shallow Reasoning: Limits of $k$NN-LMs. (4) BAM! Just Like That: Simple and Efficient Parameter Upcycling for Mixture of Experts. (5) SciDFM: A Large Language Model with Mixture-of-Experts for Science. (6) MIND: Math Informed syNthetic Dialogues for Pretraining LLMs and so on. |
| **What (other) tasks could the dataset be used for?** | MATHPILE was developed to enhance language modeling, offering significant benefits for a variety of mathematical reasoning tasks. |
| **Is there anything about the composition of the dataset or the way it was collected and preprocessed/-cleaned/labeled that might impact future uses?** | Our cleaning and filtering processes, while thorough, may not be entirely optimal, potentially leading to the exclusion of some valuable documents. Additionally, MATHPILE is specifically tailored for English, which limits its applicability in multilingual contexts. |
| **Are there tasks for which the dataset should not be used?** | Any tasks which may considered irresponsible or harmful. |

### DISTRIBUTION

| | |
|---|---|
| **Will the dataset be distributed to third parties outside of the entity on behalf of which the dataset was created?** | Yes, MATHPILE has been made available through the HuggingFace Hub (`https://huggingface.co/datasets/GAIR/MathPile`). |
| **How will the dataset will be distributed?** | MATHPILE has been made available through the HuggingFace Hub (`https://huggingface.co/datasets/GAIR/MathPile`). |
| **When will the dataset be distributed?** | The MATHPILE will be available after this paper is made public. |

| | |
|---|---|
| **Will the dataset be distributed under a copyright or other intellectual property (IP) license, and/or under applicable terms of use (ToU)?** | If the source data of MATHPILE is governed by a license more restrictive than CC BY-NC-SA 4.0, MATHPILE adheres to that stricter licensing. In all other cases, it operates under the CC BY-NC-SA 4.0 license. If any data owner objects to the use of their data, we are willing to take appropriate action immediately, including removing the relevant data. |
| **Have any third parties imposed IP-based or other restrictions on the data associated with the instances?** | Not to our knowledge. |
| **Do any export controls or other regulatory restrictions apply to the dataset or to individual instances?** | Not to our knowledge. |
| MAINTENANCE | |
| **Who will be supporting/hosting/maintaining the dataset?** | MATHPILE will be hosted on the HuggingFace Hub. |
| **How can the owner/curator/manager of the dataset be contacted?** | stefanpengfei@gmail.com   zzwang.nlp@gmail.com |
| **Is there an erratum?** | No. |
| **Will the dataset be updated?** | Yes, it is currently a work in progress and updates are ongoing. |
| **If others want to extend/augment/build on/contribute to the dataset, is there a mechanism for them to do so?** | No. |

Table 6: Datasheet for MATHPILE , following Gebru et al. [21].

# B  Ethics Statement

In the collection and creation of MATHPILE , we strictly adhered to all copyright and licensing requirements of the data sources. Specifically, we gathered a large amount of data from the internet, including mathematical textbooks, web pages, and community Q&A content, ensuring that the use of these data complies with the original licensing terms. Wikipedia, ProofWiki, and StackExchange are licensed under CC BY-SA (2.5, 3.0 or 4.0). Textbooks and arXiv are licensed under CC BY 4.0, CC BY-SA 4.0, CC BY-NC-SA 4.0 and others. Common Crawl follows the Common Crawl Foundation Terms of Use and C4 license. The final open-source MATHPILE dataset is released under the CC BY-NC-SA 4.0 license. If the source data's license is more restrictive than CC BY-NC-SA 4.0, we adopt the stricter license.

However, during the collection of some data, such as publicly available and open-source textbooks, we did not obtain explicit consent from each author. We recognize that this may involve potential copyright issues. Therefore, we have implemented the following measures to mitigate and manage these risks:

1. **Strict Selection of Data Sources**: We prioritize selecting data sources that are clearly marked with open licenses or public domain status, avoiding the use of content explicitly marked as copyright-protected or prohibited from distribution.
2. **Adherence to Fair Use Principles**: When using copyrighted and non-commercially licensed content, we adhere to the principles of fair use, aiming to promote scientific research and educational purposes rather than commercial purposes, thereby not affecting the market value of the original content.
3. **Acceptance of Feedback from Users and Content Authors**: We welcome feedback from data users and authors at any time to request the removal or modification of their data.

MATHPILE has been carefully curated and processed to minimize any potential ethical concerns. We also explicitly state that if any data owner objects to the use of their data, we are willing to take appropriate action immediately, including removing the relevant data. Through these measures, we strive to ensure the diversity and richness of the collected data while complying with relevant copyright and licensing regulations, thereby reducing potential legal risks. We bear full responsibility for any potential violations of rights or licensing issues that may arise from this dataset.

# C  Examples of MATHPILE

We provide some illustrative examples from each source in MATHPILE , as shown in Figure 4 to Figure 10.

## A document from MATHPILE-CommonCrawl

**Text:**
Are there optimizers where it is possible to specify ordinal ranking of parameters?
Assume that $f$ is smooth ($n$-th order differentiable in each of the parameters).
An approach I often use when applying unconstrained optimisation algorithms to constrained problems is to transform the parameter space such that the constraints cannot be violated.
Of course this results in $\theta_1^* \geq \theta_2^* \geq \theta_3^*$ which isn't quite what you asked for. To get a strict ranking you'll need to bump $x_1 - x_2^2$ and $x_1 - x_2^2 - x_3^2$ down at the last digit of precision.
thus spake a.k.thus spake a.k.
These variants of your constraints are linear, so provided that your function $f$ is well-behaved (smooth, easy to calculate, easy to compute derivatives, derivatives are well-conditioned, etc.), any constrained optimization solver should be able to solve your problem without issue.
Not the answer you're looking for? Browse other questions tagged optimization constrained-optimization or ask your own question.
Does the amount of correlation of model parameters matter for nonlinear optimizers?
Optimization of a blackbox function with an equality constraint?

...

**Subset**: CommonCrawl

**meta**:
    language_detection_score: 0.8670,
    char_num_after_normalized: 926,
    contain_at_least_two_stop_words: True,
    ellipsis_line_ratio: 0.0,
    idx: 383668,
    lines_start_with_bullet_point_ratio: 0.0,
    mean_length_of_alpha_words: 5.0870,
    non_alphabetical_char_ratio: 0.0,
    symbols_to_words_ratio: 0.0,
    uppercase_word_ratio: 0.0060,
    ...

Figure 4: An example Common Crawl document in MATHPILE

## A document from MATHPILE -Wikipedia

**Text:**
# Inner Automorphism

In abstract algebra, an **inner automorphism** is an automorphism of a group, ring, or algebra given by the conjugation action of a fixed element, called the *conjugating element*. They can be realized via simple operations from within the group itself, hence the adjective "inner". These inner automorphisms form a subgroup of the automorphism group, and the quotient of the automorphism group by this subgroup is defined as the outer automorphism group.

## Definition

If $G$ is a group and $g$ is an element of $G$ (alternatively, if $G$ is a ring, and $g$ is a unit), then the function

$$\varphi_g : G \to G$$
$$\varphi_g(x) := g^{-1}xg$$

is called **(right) conjugation by** $g$ (see also conjugacy class). This function is an endomorphism of $G$: for all $x_1, x_2 \in G$,

$$\varphi_g(x_1x_2) = g^{-1}x_1x_2g = (g^{-1}x_1g)(g^{-1}x_2g) = \varphi_g(x_1)\varphi_g(x_2),$$

where the second equality is given by the insertion of the identity between $x_1$ and $x_2$. Furthermore, it has a left and right inverse, namely $\varphi_{g^{-1}}$. Thus, $\varphi_g$ is bijective, and so an isomorphism of $G$ with itself, i.e., an automorphism. An **inner automorphism** is any automorphism that arises from conjugation.[1]
When discussing right conjugation, the expression $g^{-1}xg$ is often denoted exponentially by $x^g$. This notation is used because composition of conjugations satisfies the identity: $(x^{g_1})^{g_2} = x^{g_1 g_2}$ for all $g_1, g_2 \in G$. This shows that right conjugation gives a right action of $G$ on itself.

### Inner and Outer Automorphism Groups

The composition of two inner automorphisms is again an inner automorphism, and with this operation, the collection of all inner automorphisms of $G$ is a group, the inner automorphism group of $G$ denoted Inn$(G)$.
Inn$(G)$ is a normal subgroup of the full automorphism group Aut$(G)$ of $G$. The outer automorphism group, Out$(G)$, is the quotient group

$$\mathrm{Out}(G) = \frac{\mathrm{Aut}(G)}{\mathrm{Inn}(G)}.$$

The outer automorphism group measures, in a sense, how many automorphisms of $G$ are not inner. Every non-inner automorphism yields a non-trivial element of Out$(G)$, but different non-inner automorphisms may yield the same element of Out$(G)$.
Saying that conjugation of $x$ by $a$ leaves $x$ unchanged is equivalent to saying that $a$ and $x$ commute:

$$a^{-1}xa = x \iff xa = ax.$$

Therefore, the existence and number of inner automorphisms that are not the identity mapping is a kind of measure of the failure of the commutative law in the group (or ring).
An automorphism of a group $G$ is inner if and only if it extends to every group containing $G$.[2]

...

---
**Subset**: Wikipedia

---
**meta**:
    language_detection_score: 0.7236,
    char_num_after_normalized: 5794,
    contain_at_least_two_stop_words: True,
    ellipsis_line_ratio: 0.0,
    lines_start_with_bullet_point_ratio: 0.0,
    mean_length_of_alpha_words: 4.2245,
    mimetype: text/html,
    page_index: 48171,
    page_path: A/Inner_automorphism,
    page_title: Inner automorphism,
    non_alphabetical_char_ratio: 0.1422,
    symbols_to_words_ratio: 0.0,
    uppercase_word_ratio: 0.0871,
    ...

Figure 5: An example Wikipedia document in MATHPILE

**Text:**
# LINEAR TORIC FIBRATIONS

SANDRA DI ROCCO

## INTRODUCTION TO TORIC FIBRATIONS

Definition 1.1. A toric fibration is a surjective flat map $f : X \to Y$ with connected fibres where
(a) $X$ is a toric variety
(b) $Y$ is a normal algebraic variety
(c) $\dim(Y) < \dim(X)$.

Remark 1.2. Observe that if $f : X \to Y$ is a toric fibration then $Y$ and a general fiber $F$ are also toric varieties. Moreover if $X$ is smooth, respectively $\mathbb{Q}$-factorial then so is $Y$ and $F$.

Combinatorial characterization. A toric fibration has the following combinatorial characterization (see [EW, Chapter VI] for further details). Let $X = X_\Sigma$, where $\Sigma \subset N \cong \mathbb{Z}^n$, be a toric variety of dimension $n$ and let $i : \Delta \hookrightarrow N$ a sublattice.

Proposition 1.3. [EW] The inclusion $i$ induces a toric fibration if and only if:
(a) $\Delta$ is a primitive lattice, i.e. $(\Delta \otimes \mathbb{R}) \cap N = \Delta$.
(b) For every $\sigma \in \Sigma(n)$, $\sigma = \tau + \eta$, where $\tau \in \Delta$ and $\eta \cap \Delta = \{0\}$ (i.e. $\Sigma$ is a splitfan).

We briefly outline the construction. The projection $\pi : N \to N/\Delta$ induces a map of fans $\Sigma \to \pi(\Sigma)$ and thus a map of toric varieties $f : X \to Y$. The general fiber $F$ is a toric variety defined by the fan $\Sigma_F = \{\sigma \in \Sigma \cap \Delta\}$.

When the toric variety $X$ in a toric fibration is polarized by an ample line bundle $L$ we will call the pair $(f : X \to Y, L)$ a polarized toric fibration. Observe that the polarized toric varieties $(X, L)$ and $\left(F, L|_F\right)$, for a general fiber $F$, define lattice polytopes $P_{(X,L)}$, $P_{\left(F, L|_F\right)}$. The polytope $P_{(X,L)}$ is in fact a "twisted sum" of a finite number of lattice polytopes fibering over $P_{\left(F, L|_F\right)}$. Definition 1.4. Let $R_0, \ldots, R_k \subset \Delta$ be polytopes. Let $\pi : M \to \Lambda$ be a surjective map of lattices such that $\pi(R_i) = v_i$ and the $v_0, \cdots, v_k$ are distinct vertices of $\mathrm{Conv}(v_0, \ldots, v_k)$. We will call a Cayley $\pi$-twisted sum (or simply a Cayley sum) of $R_0, \ldots, R_k$ a polytope which is affinely isomorphic to $\mathrm{Conv}(R_0, \ldots, R_k)$. We will denote it by:

$$[R_0 \star \ldots \star R_k]_\pi$$

If the polytopes $R_i$ are additionally normally equivalent, i.e. they define the same normal fan $\Sigma_Y$, we will denote the Cayley sum by:

$$\mathrm{Cayley}\ (R_0, \ldots, R_k)_{(\pi, Y)}.$$

These are the polytopes that are associated to a polarized toric fibration. Consider a sublattice $i : \Delta \hookrightarrow N$ and the dual lattice surjection $\pi : M \to \Lambda$.

Proposition 1.5. [CDR08] The sublattice $i : \Delta \hookrightarrow N$ induces a polarized toric fibration $(f : X \to Y, L)$ if and only if $P_{(X,L)} = \mathrm{Cayley}(R_0, \ldots, R_k)_{(\pi, Y)}$ for some normally equivalent polytopes $R_0, \ldots, R_k$.

The polarized general fiber $\left(F, L|_F\right)$ corresponds to the polarized toric variety associated to the polytope $P_{\left(F, L|_F\right)} = \mathrm{Conv}(v_0, \ldots, v_k)$ and the polytopes $R_0, \cdots, R_k$ define the embeddings of the invariant sections polarized by the restrictions of $L$.

Example 1.6. Consider the Hirzebruch surface $\mathbb{F}_1 = Bl_p\left(\mathbb{P}^2\right) = \mathbb{P}\left(\mathscr{O}_{\mathbb{P}1} \oplus \mathscr{O}_{\mathbb{P}1}(1)\right)$ polarized by the tautological line bundle $\xi = 2\phi^*\left(\mathscr{O}_{\mathbb{P}2}(1)\right) - E$ where $\phi$ is the blow-up map and $E$ the exceptional divisor. The associated polytope is $P = \mathrm{Cayley}(\Delta_1, 2\Delta_1)$.

FIGURE 1. The Hirzebruch surface $\mathbb{P}\left(\mathscr{O}_{\mathbb{P}1} \oplus \mathscr{O}_{\mathbb{P}1}(1)\right)$

Example 1.7. More generally:
- when $\pi(P) = \Delta_t$ the polytope $\mathrm{Cayley}(R_0, \ldots, R_k)_{(\pi, Y)}$ defines the variety $\mathbb{P}(L_0 \oplus \ldots \oplus L_k)$, where the $L_i$ are ample line bundles on the toric variety $Y$, polarized by the tautological bundle $\xi$. In particular $L|_F = \mathscr{O}_{\mathbb{P}t}(1)$.
- When $\pi(P)$ is a simplex (not necessarily smooth) $\mathrm{Cayley}(R_0, \ldots, R_k)_{(\pi, Y)}$ defines a Mori-type fibration. A fibration whose general fiber has Picard rank one. - When $\pi(P) = s\Delta_t$ then again the variety has the structure of a $\mathbb{P}^t$-fibration whose general fiber $F$ is embedded via an $s$-Veronese embedding: $\left(F, L|_F\right) = \left(\mathbb{P}^t, \mathscr{O}_{\mathbb{P}t}(s)\right)$.

For general Cayley sums, $[R_0 \star \ldots \star R_k]_\pi$, one has the following geometrical interpretation. Let $(X, L)$ be the associated polarized toric variety and let $Y$ be the toric variety defined by the Minkowski sum $R_0 + \ldots + R_k$. The fan defining $Y$ is a refinement of the normal fan of $R_i$ for $i = 0, \ldots, k$. Consider the associated birational maps $\phi_i : Y \to Y_i$, where $(Y_i, L_i)$ is the polarized toric variety defined by the polytope $R_i$. The line bundles $H_i = \phi_i^*(L_i)$ are nef line bundles on $Y$. Denote by the same symbol the maps of fans $\phi_i : \Sigma_Y \to \Sigma_{Y_i}$. Define then the fan:

$$\Sigma_Z : \left\{\phi_i^{-1}(\sigma_j) \times \eta_l, \text{ for all } \sigma_j \in \Sigma_{Y_i}, \eta_l \in \Sigma_\Delta\right\}$$

where $\Lambda = \mathrm{Conv}(v_0, \ldots, v_k)$. It is a refinement of $\Sigma_X$ and thus the defining variety $Z$ is birational to $X$. Moreover it is a split fan and thus it defines a toric fibration $f : Z \to Y$. The Cayley sum $[R_0 \star \ldots \star R_k]_\pi$ is the polytope defined by the nef line bundle $\phi^*(L)$, and the polytopes $R_i$ are the polytopes defined by the nef line bundles $H_i$ on the invariant sections.

Historical Remark. The definition of a Cayley polytope originated by what is "classically" referred to as the Cayley trick. We first recall the definition of Resultant and Discriminant. Let $f_1(x), \ldots, f_n(x)$ be a system of $n$ polynomials in $n$ variables $x = (x_1, \ldots, x_n)$ supported on $A \subset \mathbb{Z}^n$. This means that $f_i = \Pi_{a_j \in A} c_j x^{d_j}$. The resultant (of $A$), $R_A(c_j)$, is a polynomial in the coefficients $c_j$, which vanishes whenever the corresponding polynomials have a common zero.

The discriminant of a finite subset $A$, $\Delta_\mathscr{A}$, is also a polynomial $\Delta_\mathscr{A}(c_j)$ in the variables $c_j \in A$ which vanishes whenever the corresponding polynomial has a multiple root.

Theorem 1.8. [GKZ][Cayley Trick] The A-resultant of the system $f_1, \ldots, f_n$ equals the Adiscriminant of the polynomial:

$$p(x, y) = f_i(x) + \sum_2^n y_{i-1} f_i(x).$$

Let $R_i = N(f_i) \subset \mathbb{R}^n$ be the Newton polytopes of the polynomials $f_i$. The Newton polytope of the polynomial $p(x, y)$ is the Cayley sum $[R_1 \star \ldots \star R_n]_\pi$, where $\pi : \mathbb{R}^{2n-1} \to \mathbb{R}^{n-1}$ is the natural projection such that $\pi\left([R_1 \star \ldots \star R_n]_\pi\right) = \Delta_{n-1}$.

...

---

**Subset**: Textbooks

---

**meta**:
    book_name: Linear Toric Fibrations_Sandra Di Rocco,
    type: Notes,
    ...

Figure 6: An example textbook document in MATHPILE

## A document from MATHPILE-ProofWiki

**Text:**
\section{Test for Submonoid}
Tags: Abstract Algebra, Monoids

\begin{theorem}
To show that \struct {T, circ} is a submonoid of a monoid \struct {S, circ}, we need to show that:
:(1):    $T \subseteq S$
:(2):    \struct {T, circ} is a magma (that is, that it is closed)
:(3):    \struct {T, circ} has an identity.
\end{theorem}

\begin{proof}
From Subsemigroup Closure Test, $(1)$ and $(2)$ are sufficient to show that \struct {T, circ} is a subsemigroup of \struct {S, circ}.
Demonstrating the presence of an identity is then sufficient to show that it is a monoid. {{qed}}
Category:Monoids
\end{proof}

...

**Subset**: ProofWiki

**meta**:
    type: Theorem_Proof,
    ...

Figure 7: An example ProofWiki (a theorem and its proof) document in MATHPILE

## A document from MATHPILE-ProofWiki

**Text:**
\begin{definition}[Definition:That which produces Medial Whole with Medial Area/Whole]

Let $a, b \in \mathcal{R}_{>0}$ be (strictly) positive real numbers such that $a > b$.

Let $a - b$ be a straight line which produces with a medial area a medial whole.

The real number $a$ is called the "'whole"' of the straight line which produces with a medial area a medial whole.

Category:Definitions/Euclidean Number Theory
\end{definition}

**Subset**: ProofWiki

**meta**:
    type: Definition,
    ...

Figure 8: An example ProofWiki (definition) document in MATHPILE

## A document from MATHPILE-arXiv

**Text:**
\begin{document}
\title{Coherence freeze in an optical lattice investigated via pump-probe spectroscopy}
\author{Samansa Maneshi}
\email[]{smaneshi@physics.utoronto.ca}
\author{Chao Zhuang}
\author{Christopher R. Paul}
\author{Luciano S. Cruz}
\altaffiliation[Current address: ]{UFABC, São Paulo, Brazil.}
\author{Aephraim M. Steinberg}
\affiliation{Centre for Quantum Information & Quantum Control and Institute for Optical Sciences,
Department of Physics, University of Toronto, Canada }
\date{\today}
\pacs{37.10.Jk, 03.65.Yz, 03.67.-a, 42.50.Md}

\begin{abstract}
Motivated by our observation of fast echo decay and a surprising coherence freeze, we have developed a pump-probe spectroscopy technique for vibrational states of ultracold $^{85}$Rb atoms in an optical lattice to gain information about the memory dynamics of the system. We use pump-probe spectroscopy to monitor the time-dependent changes of frequencies experienced by atoms and to characterize the probability distribution of these frequency trajectories. We show that the inferred distribution, unlike a naive microscopic model of the lattice, correctly predicts the main features of the observed echo decay.
\end{abstract}

\maketitle

Characterizing decoherence mechanisms is a crucial task for experiments aiming to control quantum systems, e.g., for quantum information processing (QIP). In this work, we demonstrate how two-dimensional (2D) pump-probe spectroscopy may be extended to provide important information on these mechanisms. As a model system, we study quantum vibrational states of ultracold atoms in an optical lattice. In addition to being a leading candidate system for QIP \citeBrennenJaksch, optical lattices are proving a versatile testing ground for the development of quantum measurement and control techniques \citeOMandel, Anderlini and a powerful tool for quantum simulations, e.g. the study of Anderson localization and the Hubbard model \citeMottAnderson.

In our experiment, we study the vibrational coherence of $^{85}$Rb atoms trapped in a shallow one-dimensional standing wave. Through our 2D pump-probe technique, we obtain detailed microscopic information on the frequency drift experienced by atoms in the lattice, enabling us to predict the evolution of coherence. Since the pioneering development of the technique in NMR\citeJeener-Ernst, 2D spectroscopy has been widely used to obtain high-resolution spectra and gain information about relaxations, couplings, and many-body interactions, in realms ranging from NMR \citeErnst to molecular spectroscopy \citeMukamel-Jonas, Hybl, Brixner, MillerNature to semiconductor quantum wells \citeCundiff, KWStone. Here, we show that similar powerful techniques can be applied to the quantized center-of-mass motion of trapped atoms, and more generally, offer a new tool for the characterization of systems in QIP and quantum control.

\begin{figure}
\caption{(Color online) Two typical measurements of echo amplitude vs. time. The echo pulse and the observed echo envelope are centered at times $t_p$ and $2t_p$, respectively. After an initial decay, echo amplitude stays constant for about $1ms$ forming a plateau, before decaying to zero. The average lattice depths are $20E_R$ (circles) and $18E_R$ (squares).}
\label{fig1}
\end{figure}

We have previously measured the evolution of coherence between the lowest two vibrational states of potential wells \cite{Ours}.

The dephasing time is about $0.3ms$ ($T_2^\star$).

This dephasing is partly due to an inhomogeneous distribution of lattice depths as a result of the transverse Gaussian profile of the laser beams. To measure the homogeneous decoherence time ($T_2$), we perform pulse echoes, measuring the echo amplitude as a function of time \cite{Ours}.

Figure \ref{fig1} shows two typical measurements of echo amplitude carried out on different dates under slightly different conditions such as different average lattice depths and different dephasing times. The echo amplitude initially decays with a time constant of about $0.7ms$, which is much faster than the photon scattering time ($\sim 60ms$) in the lattice. It then exhibits a $1ms$-long coherence freeze followed by a final decay. Absent real decoherence on the short time scale of $1ms$, only loss of frequency memory would inhibit the appearance of echoes. This loss comes about when atoms experience time-varying frequencies. We use 2D pump-probe spectroscopy to monitor this frequency drift. Our 2D pump-probe spectroscopy is essentially a version of spectral hole-burning for vibrational states. By monitoring the changes in the hole spectrum as a function of time we gain information on the atoms' frequency drift.

Information obtained from our 2D spectra enables us to characterize the temporal decay of frequency memory and through our simulations we find that "coherence freeze" is related to the shape of this memory loss function.

Similar plateaus in echo decay and a two-stage decay of echo amplitude have been observed in a Cooper-pair box \cite{Nakamura}, for a single electron spin in a quantum dot \cite{Vandersypen} and for electron spins in a semiconductor \cite{SClark}. Those plateaus or two-stage decays have been either explained through {\it{a priori}} models or simply described phenomenologically. Here, we are introducing an experimental technique to directly probe the origin of plateaus.

The periodic potential in our experiment is formed by interfering two laser beams blue-detuned by $25GHz$ from the D2 transition line, $F = 3 \rightarrow F' = 4$ ($\lambda = 780nm$), thus trapping atoms in the regions of low intensity, which minimizes the photon scattering rate and the transverse forces. The two laser beams intersect with parallel linear polarizations at an angle of $\theta = (49.0 \pm 0.2)^\circ$, resulting in a spacing of $L = (0.930 \pm 0.004)\mu m$ between the wells. Due to gravity, the full effective potential also possesses a "tilt" of $2.86E_R$ per lattice site, where $E_R = \frac{h^2}{8mL^2}$ is the effective lattice recoil energy. The photon scattering time in our experiment is $\approx 60ms$ and the Landau-Zener tunneling times for transitions from the lowest two levels are greater than $160ms$. Atoms are loaded into the lattice during a molasses cooling stage and prepared in the ground vibrational state by adiabatic filtering \cite{StefanQPT}. Due to the short coherence length of atoms in optical molasses ($60nm$ at $10\mu K$), there is no coherence between the wells. We measure populations of atoms in the ground vibrational, the first excited, and the (lossy) higher excited states $P_1$, $P_2$, and $P_L$, respectively, by fluorescence imaging of the atomic cloud after adiabatic filtering \cite{StefanQPT}.

...

**Subset**: arXiv

**meta**:
    id: 1005.2635,
    language_detection_score: 0.8389,
    ...

Figure 9: An example arXiv document in MATHPILE

**Question:**

Title: Are fractions hard because they are like algebra?

Body:

It occurs to me that to really understand the ways that people work with fractions on paper requires a good grasp of the ideas that numbers have multiple representations and that expressions can be manipulated in various ways without changing the number they represent. These are essentially algebraic ideas.

For example, adding fractions requires us to rewrite the fractions in a different form, and then essentially factorise the expression. This is the same as rearranging expressions in algebra. Dividing fractions requires us to rerepresent an operation like $\div \frac{2}{3}$ as $\times \frac{3}{2}$. This is the same as realising the connection between operations that you use to solve equations in algebra. And cancelling down before multiplying is very sophisticated rewriting relying on various associative and commutative laws.

So it seems that we are really asking children to think in algebraic ways in order to understand fraction calculations well. This would seem to me to be a good reason why children and adults find it hard - they need more scaffolding in some abstract ideas.

Is this a reasonable theory and has anyone written about this algebra-fractions connection before? To be clear, I am not asking if this is the only reason fractions are hard, but if there is any discussion out there to draw parallels between learning algebra and learning to manipulate fractions.

Id: 7826

Score: 17

Tags: <algebra><fractions>

LastEditorDisplayName: None

OwnerDisplayName: None

ClosedDate: None

FavoriteCount: None

language_detection_score: 0.9558

...

**Answers**:

Body: Not sure about paper references. One reason why people don´t understand fractions is because they are seemingly illogical.

You score one basket out of three 1/3.

A little while later you try again and score 1/2. Clearly you have scored 2/5 shots? In many ways this is the correct answer. So why shouldn´t $\frac{1}{3} + \frac{1}{2} = \frac{2}{5}$

People generally don´t understand equivalent fractions. It is strange for one farmer to say there are 4 sheep and another to say there are 8/2 sheep in the same field. People assume that the number 4 does what it says on the tin and is how we always describe 4 ness of something. They don´t understand equivalence.

Partly to blame is treating fractions like conjuring tricks. If this is the question...do this, if this is the question ...do another uncorrelated thing. I asked my class (who seemingly could compute $\frac{2}{3} \times \frac{3}{5}$ correctly) to draw me a picture

instead of just multiplying. No one could do it yet they all said "but it´s $frac615$ you times the top and the bottom!"

I think drawing fractions is extremely useful. Draw $\frac{2}{3} \div 2$ or $2 \div \frac{2}{3}$ It´s not easy but I find students develop robustness eventually and begin to abstract themselves.

Id: 7827,

Score: 9,

is_accepted_answer: False,

language_detection_score: 0.9599,

Body: The obvious (to me) source of difficulty is that fractions are just plain complicated, more so than almost anything else in elementary education. You have to operate with a pair of numbers, instead of a single one, and you have to keep the order straight. Adding is quite complicated in its own right. Things are further complicated by rules about least common denominators and least terms.

I´m a little unclear about the question´s emphasis on algebra. Any sort of general rule or operation in arithmetic must have a connection to algebra, but I do not see what is intrinsically difficult about algebra that relates to numeric fractions. Certainly some parts of algebra are hard, and some parts harder than others, algebraic fractions among them. It seems to me that fractions are difficult because it´s easy to confuse the various bits. Even when you´ve got them straight, they´re noticeably slower to use, take concentration, and when things have such cognitive demands, they´re harder to think with.

Conceptually, they´re a little bit odd, which is probably distracting until you get used to them. What they represent do not seem to apply to the same things that (whole) numbers do. Evidently fractions are not considered in this passage:

In that city, which was the oldest in the world, the cat was an object of veneration. Its worship was the religion of the country. The multiplication and addition of cats were a perpetual instruction in arithmetic. Naturally, any inattention to the wants of a cat was punished with great severity in this world and the next... – A. Bierce, "A Revolt of the Gods"

Now to have one-and-a-half cats seems a very different thing than to have three halves. In the former case, there´s a good chance that the one cat you have will be alive and purring, while the same could not possibly be said about any of the halves. No doubt such lessons are considered blasphemous in that city. While many things may be divided into parts – cars are a better example than cats – not many can be divided into equivalent parts that can be used as a basis for fractions. As we get used to fractions, as well as real numbers, we are taught to ignore this and accept statements such as "the average family has 2.4 children." Here is another example:

By then, she will have shed 80 of the 240 pounds she weighed in with when she entered Peter Bent Brigham hospital obesity program. A third of her left behind! – The Boston Herald American, 7/7/77

The question seems to welcome references. There are certainly several that connect fractions with algebra. This paper,

Seigler et al. (2013), Fractions: the new frontier for theories of numerical development, Trends in Cognitive Sciences,

is a short survey of what is known and unknown about neural bases for one´s knowledge of fractions. Whole number arithmetic knowledge has been studied, and the authors suggest that the representation of the knowledge fractions is an area ripe for investigation. It reviews (with references) why fractions are difficult and the relation of skill at fractions to skill at algebra. Generally – or, rather, I only know of papers that discuss the connection in that direction, with algebra skill being dependent on fractions skill. (OTOH, I´m not widely read in this area.)

Id: 7831,

Score: 11,

is_accepted_answer: False,

language_detection_score: 0.9780

**Subset**: StackExchange

Figure 10: An example StackExchange document in MATHPILE . Here is a question from "matheducators" ".stackexchang.com" with two high-quality responses.

# D Details for Corpus Collection and Processing

The subjects from which we collected papers on arXiv are listed in Table 7. The specific StackExchange sites from which we gathered data are listed in Table 8. We illustrate the LaTeX display issue with an example in Figure 11.

During the collection process of arXiv, we undertook extensive transformations to enhance data clarity and consistency. Specifically, we (1) removed comments in each paper; (2) reverted many macro commands (e.g., "newcommand") to their original forms; (3) omitted figure environments while retaining captions and figure labels; (4) excluded acknowledgements sections; (5) eliminated references in each paper; (6) condensed more than three consecutive empty lines to two; (7) replaced certain formatting commands like "hfill" and "vspace" with an empty line; (8) replaced the "maketitle" command in the main document body with the actual title (if available); (9) preserved only the content within the main body of the LaTex document.

We summarize the parts of the dataset collection (cf. § 2) and global data preprocessing (cf. § 3) where human intervention was involved and whether the cleaning process was automated in the Table 9 and Table 10. We hope this provides a clearer understanding of MATHPILE construction process.

Table 7: The subject list during collecting corpus from arXiv.

| Subjects |
| --- |
| math.AG, math.AT, math.AP, math.CT, math.CA, math.CO, math.AC, math.CV, math.DG, math.DS, math.FA, math.GM, math.GN, math.GT, math.GR, math.HO, math.IT, math.KT, math.LO, math.MP, math.MG, math.NT, math.NA, math.OA, math.OC, math.PR, math.QA, math.RT, math.RA, math.SP, math.ST, math.SG, math-ph, quant-ph, cs.CC, cs.CG, cs.DM, cs.DS, cs.FL, cs.GT, cs.LG, cs.NA, cs.LO, q-fin.MF, stat.CO, stat.ML, stat.ME, stat.OT, stat.TH, econ.TH |

Table 8: The site list during collecting corpus from StackExchange.

| Sites sourced from StackExchange |
| --- |
| math.stackexchange.com, mathoverflow.net, mathematica.stackexchange.com, matheducators.stackexchange.com, hsm.stackexchange.com, physics.stackexchange.com, proofassistants.stackexchange.com, tex.stackexchange.com, datascience.stackexchange, cstheory.stackexchange.com, cs.stackexchange.com |

Table 9: Details of Human Involvement and Automation in the MATHPILE Collection Process

| MATHPILE Subset | Human Involvement in Data Collection | Cleaning Process Automated? |
|---|---|---|
| Textbooks | Manual search and download of open-source, free mathematics textbooks; quality check; setting cleaning rules | Yes, the automated application of the PDF conversion API and the document cleaning rules |
| arXiv Papers | Manual selection of relevant mathematical field categories; setting latex cleaning and formatting rules | Yes, automated cleaning steps like comment removal and format conversion |
| Wikipedia Mathematical Entries | Humans observed samples to define rules for cleaning irrelevant content such as copyright statements | Yes, automated HTML to Markdown conversion; and removal of extraneous lines |
| ProofWiki Entries | No significant human intervention (mainly data dump selection, reformatting design) | Yes, automated text parsing and formatting |
| StackExchange Discussions | Selection of relevant mathematics-related sites within the StackExchange network; setting filter thresholds | Yes, automated HTML parsing and conversion, score filtering |
| Common Crawl Web Pages | Manual adjustment of TF-IDF rules to improve mathematical content identification | Yes, automated application of rule-based mathematical document filtering |

Table 10: Details of Human Involvement and Automation in the MATHPILE Global Data Processing Steps

| Global Data Processing Step | Human Involvement | Cleaning Process Automated? |
|---|---|---|
| Language Identification | Manual adjustment of thresholds based on observed false positives | Yes, using FastText with custom thresholds |
| Data Cleaning & Filtering | Human observation to define rules for filtering irrelevant content | Yes, automated application of rules for content filtering and removal |
| Data Deduplication | Manual review of near-duplicate samples from different sources | Yes, automated using MinHash LSH |
| Data Contamination Detection | Human verification of flagged benchmark duplicates | Yes, automated detection based on pre-defined criteria |

Figure 11: A document processed by `html2text` (above) compared to one obtained through another library **Resiliparse** plus DOM parsing (below).

# E   Example for Quality Annotation

We present a cleaned example document with quality annotations (see Figure 12).

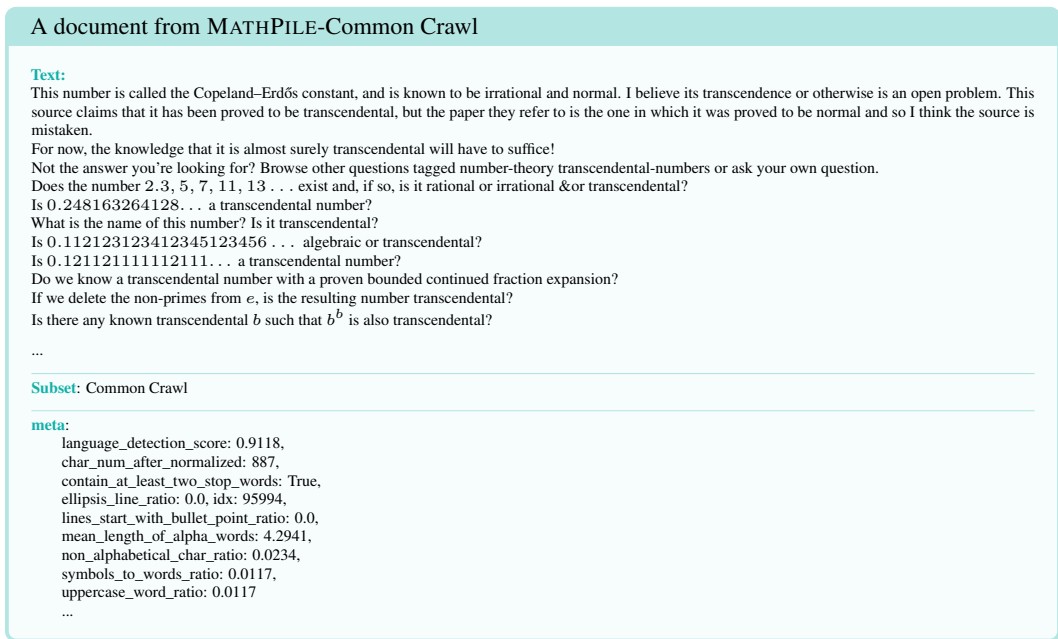

Figure 12: An example document after cleaning and filtering with quality annotations

# F   Examples of Duplicates Encountered in the Deduplication Process

We provide some illustrative examples of duplicates from each source in the deduplication process, as shown in Table 11 to Table 16.

We also provide examples of downstream task benchmarks (i.e., MATH and MMLU-STEM) leaks identified during our data contamination detection process for our corpus (as shown in Table 17 and Table 18) and OpenWebMath (as shown in Table 19).

Table 11: Near-duplication matches found in CommonCrawl by MinHash LSH deduplication (in *italics*).

| | |
|---|---|
| *In algebraic topology we often encounter chain complexes with extra multiplicative structure. For example, the cochain complex of a topological space has what is called the $E_\infty$-algebra structure which comes from the cup product.* *In this talk I present an idea for studying such chain complexes, $E_\infty$ differential graded algebras ($E_\infty$ DGAs), using stable homotopy theory. Namely, I discuss new equivalences between $E_\infty$ DGAS that are defined using commutative ring spectra.* ring spectra are equivalent. *Quasi-isomorphic $E_\infty$ DGAs are $E_\infty$ topologically equivalent. However, the examples I am going to present show that the opposite is not true; there are $E_\infty$ DGAs that are $E_\infty$ topologically equivalent but not quasi-isomorphic. This says that between $E_\infty$ DGAs, we have more equivalences than just the quasi-isomorphisms. I also discuss interaction of $E_\infty$ topological equivalences with the Dyer-Lashof operations and cases where $E_\infty$ topological equivalences and quasi-isomorphisms agree.* | Özet : *In algebraic topology we often encounter chain complexes with extra multiplicative structure. For example, the cochain complex of a topological space has what is called the $E_\infty$-algebra structure which comes from the cup product. In this talk I present an idea for studying such chain complexes, $E_\infty$ differential graded algebras ($E_\infty$ DGAs), using stable homotopy theory. Namely, I discuss new equivalences between $E_\infty$ DGAS that are defined using commutative ring spectra.* We say $E_\infty$ DGAs are $E_\infty$ topologically equivalent when the corresponding commutative ring spectra are equivalent. *Quasi-isomorphic $E_\infty$ DGAs are $E_\infty$ topologically equivalent. However, the examples I am going to present show that the opposite is not true; there are $E_\infty$ DGAs that are $E_\infty$ topologically equivalent but not quasi-isomorphic. This says that between $E_\infty$ DGAs, we have more equivalences than just the quasi-isomorphisms. I also discuss interaction of $E_\infty$ topological equivalences with the Dyer-Lashof operations and cases where $E_\infty$ topological equivalences and quasi-isomorphisms agree.* |
| Université de la Saskatchewan, 1 - 4 juin 2015 www.smc.math.ca//2015f Comité d'organisation Financement étudiants Minisymposia invités Minisymposia libres Conférences libres Horaire - Minisymposa invités Open Problems Graphs and matrices Responsable et président: Shaun Fallat et Karen Meagher (University of Regina) *WAYNE BARRETT, Brigham Young University* *The Fielder Vector and Tree Decompositions of Graphs [PDF]* *In the 1970's Fiedler initiated a study of the second smallest eigenvalue of the Laplacian matrix $L$ of a graph and the corresponding eigenvector(s). These "Fiedler" vectors have become spectacularly successful in revealing properties of the associated graph. A tree decomposition $\mathcal{T}$ of a graph $G = (V, E)$ is an associated tree whose nodes are subsets of $V$ and whose edge set respects the structure of $G$. Tree decompositions have been used in the analysis of complex networks. This talk reports on an algorithm developed by students at BYU for obtaining a tree decomposition by means of Fiedler vector(s) of $G$.* *...* *Graphs that have a weighted adjacency matrix with spectrum $\{\lambda_1^{n-2}, \lambda_2^2\}$ [PDF]* *In this talk I will characterize the graphs which have an edge weighted adjacency matrix belonging to the class of $n \times n$ involutions with spectrum equal to $\{\lambda_1^{n-2}, \lambda_2^2\}$ for some $\lambda_1$ and some $\lambda_2$. The connected graphs turn out to be the cographs constructed as the join of at least two unions of pairs of complete graphs, and possibly joined with one other complete graph.* | University of Saskatchewan, June 1 - 4, 2015 www.cms.math.ca//2015 Invited Minisymposia Contributed Minisymposia Contributed Talks Graphs and matrices Organizer and Chair: Shaun Fallat and Karen Meagher (University of Regina) *WAYNE BARRETT, Brigham Young University* *The Fielder Vector and Tree Decompositions of Graphs [PDF]* *In the 1970's Fiedler initiated a study of the second smallest eigenvalue of the Laplacian matrix $L$ of a graph and the corresponding eigenvector(s). These "Fiedler" vectors have become spectacularly successful in revealing properties of the associated graph. A tree decomposition $\mathcal{T}$ of a graph $G = (V, E)$ is an associated tree whose nodes are subsets of $V$ and whose edge set respects the structure of $G$. Tree decompositions have been used in the analysis of complex networks. This talk reports on an algorithm developed by students at BYU for obtaining a tree decomposition by means of Fiedler vector(s) of $G$.* *...* *Graphs that have a weighted adjacency matrix with spectrum $\{\lambda_1^{n-2}, \lambda_2^2\}$ [PDF]* *In this talk I will characterize the graphs which have an edge weighted adjacency matrix belonging to the class of $n \times n$ involutions with spectrum equal to $\{\lambda_1^{n-2}, \lambda_2^2\}$ for some $\lambda_1$ and some $\lambda_2$. The connected graphs turn out to be the cographs constructed as the join of at least two unions of pairs of complete graphs, and possibly joined with one other complete graph.* |

Table 12: A near-duplication match found in arXiv by MinHashLSH deduplication (in *italics*).

| | |
|---|---|
| \begin{document} | \begin{document} |
| \title{Querying Guarded Fragments via Resolution} | \title{Limited memory Kelley's Method Converges for Composite Convex and Submodular Objectives} |
| \section{A detailed example} | \section{A detailed example} |
| Here we include some equations and theorem-like environments to show how these are labeled in a supplement and can be referenced from the main text. Consider the following equation: \begin{equation} \label{eq:suppa} a^2 + b^2 = c^2. \end{equation} You can also reference equations such as \cref{eq:matrices,eq:bb} from the main article in this supplement. \lipsum[100-101] | Here we include some equations and theorem-like environments to show how these are labeled in a supplement and can be referenced from the main text. Consider the following equation: \begin{equation} \label{eq:suppa} a^2 + b^2 = c^2. \end{equation} You can also reference equations such as \cref{eq:matrices,eq:bb} from the main article in this supplement. \lipsum[100-101] |
| \begin{theorem} An example theorem. \end{theorem} | \begin{theorem} An example theorem. \end{theorem} |
| \lipsum[102] | \lipsum[102] |
| \begin{lemma} An example lemma. \end{lemma} | \begin{lemma} An example lemma. \end{lemma} |
| \lipsum[103-105] | \lipsum[103-105] |
| Here is an example citation: \cite{KoMa14}. | Here is an example citation: \cite{KoMa14}. |
| \section[Proof of Thm]{Proof of \cref{thm:bigthm}} \label{sec:proof} | \section[Proof of Thm]{Proof of \cref{thm:bigthm}} \label{sec:proof} |
| \lipsum[106-112] | \lipsum[106-112] |
| \section{Additional experimental results} \Cref{tab:foo} shows additional supporting evidence. | \section{Additional experimental results} \Cref{tab:foo} shows additional supporting evidence. |
| \begin{table}[htbp] {\footnotesize \caption{Example table} \label{tab:foo} \begin{center} \begin{tabular}{\|c\|c\|c\|} \hline Species & \bf Mean & \bf Std.~Dev. \\\hline 1 & 3.4 & 1.2 \\ 2 & 5.4 & 0.6 \\\hline \end{tabular} \end{center} } \end{table} | \begin{table}[htbp] {\footnotesize \caption{Example table} \label{tab:foo} \begin{center} \begin{tabular}{\|c\|c\|c\|} \hline Species & \bf Mean & \bf Std.~Dev. \\\hline 1 & 3.4 & 1.2 \\ 2 & 5.4 & 0.6 \\\hline \end{tabular} \end{center} } \end{table} |
| \end{document} | \end{document} |

\section{Definition:Constructed
    Semantics/Instance 4/Rule of
    Idempotence}
Tags: Formal Semantics

\begin{theorem}
The Rule of Idempotence:
:$(p \lor p) \implies p$
is a tautology in Instance 4 of
    constructed semantics.
\end{theorem}

\begin{proof}
By the definitional abbreviation for
    the conditional:
:$\mathbf A \implies \mathbf B =_{\
    text{def}} \neg \mathbf A \lor \
    mathbf B$
the Rule of Idempotence can be
    written as:
: $\neg \left({p \lor p}\right) \lor
    p$
This evaluates as follows:
:$\begin{array}{|cccc|c|c|} \hline
\neg & (p & \lor & p) & \lor & p \\
\hline
1 & 0 & 0 & 0 & 0 & 0 \\
0 & 1 & 1 & 1 & 0 & 1 \\
0 & 2 & 2 & 2 & 0 & 2 \\
2 & 3 & 3 & 3 & 0 & 3 \\
\hline
\end{array}$
{{qed}}
Category:Formal Semantics
\end{proof}

\section{Definition:Constructed
    Semantics/Instance 5/Rule of
    Idempotence}
Tags: Formal Semantics

\begin{theorem}
The Rule of Idempotence:
:$(p \lor p) \implies p$
is a tautology in Instance 5 of
    constructed semantics.
\end{theorem}

\begin{proof}
By the definitional abbreviation for
    the conditional:
:$\mathbf A \implies \mathbf B =_{\
    text{def}} \neg \mathbf A \lor \
    mathbf B$
the Rule of Idempotence can be
    written as:
: $\neg \left({p \lor p}\right) \lor
    p$
This evaluates as follows:
:$\begin{array}{|cccc|c|c|} \hline
\neg & (p & \lor & p) & \lor & p \\
\hline
1 & 0 & 0 & 0 & 0 & 0 \\
0 & 1 & 1 & 1 & 0 & 1 \\
3 & 2 & 2 & 2 & 0 & 2 \\
0 & 3 & 3 & 3 & 0 & 3 \\
\hline
\end{array}$
{{qed}}
Category:Formal Semantics
\end{proof}

\section{Imaginary Part of Complex
    Product}
Tags: Complex Multiplication

\begin{theorem}
Let $z_1$ and $z_2$ be complex
    numbers.
Then:
:$\map \Im {z_1 z_2} = \map \Re {z_1
    } \, \map \Im {z_2} + \map \Im {
    z_1} \, \map \Re {z_2}$
\end{theorem}

\begin{proof}
Let $z_1 = x_1 + i y_1$ and $z_2 =
    x_2 + i y_2$.
By definition of complex
    multiplication:
:$z_1 z_2 = x_1 x_2 - y_1 y_2 + i \
    paren {x_1 y_2 + x_2 y_1}$
Then
{{begin-eqn}}
{{eqn | l = \map \Im {z_1 z_2}
     | r = x_1 y_2 + x_2 y_1
     | c = {{Defof|Imaginary Part}}
}}
{{eqn | r = \map \Re {z_1} \, \map \
    Im {z_2} + \map \Im {z_1} \, \map
     \Re {z_2}
     | c = {{Defof|Imaginary Part}}
}}
{{end-eqn}}
{{qed}}
\end{proof}

\section{Real Part of Complex
    Product}
Tags: Complex Multiplication

\begin{theorem}
Let $z_1$ and $z_2$ be complex
    numbers.
Then:
:$\map \Re {z_1 z_2} = \map \Re {z_1
    } \map \Re {z_2} - \map \Im {z_1}
     \map \Im {z_2}$
\end{theorem}

\begin{proof}
Let $z_1 = x_1 + i y_1$ and $z_2 =
    x_2 + i y_2$.
By definition of complex
    multiplication:
:$z_1 z_2 = x_1 x_2 - y_1 y_2 + i \
    paren {x_1 y_2 + x_2 y_1}$
Then:
{{begin-eqn}}
{{eqn | l = \map \Re {z_1 z_2}
     | r = x_1 x_2 - y_1 y_2
     | c = {{Defof|Real Part}}
}}
{{eqn | r = \map \Re {z_1} \map \Re
    {z_2} - \map \Im {z_1} \map \Im {
    z_2}
     | c = {{Defof|Real Part}}
}}
{{end-eqn}}
{{qed}}
\end{proof}

Table 14: Duplication matches found in Wikipedia by MinHash LSH deduplication (in *italics*).

*# HP-42S*

*The \*\*HP-42S RPN Scientific\*\* is a programmable RPN Scientific hand held calculator introduced by Hewlett-Packard in 1988. It has advanced functions suitable for applications in mathematics, linear algebra, statistical analysis, computer science and others.*

*HP-42S*
*The HP-42S*
*—*
*Type| Programmable scientific*
*Manufacturer| Hewlett-Packard*
*Introduced| 1988*
*Discontinued| 1995 Calculator*
*Entry mode| RPN*
*Precision| 12 display digits (15 digits internally),[1] exponent ±499*
*Display type| LCD dot-matrix*
*Display size| 2 lines, 22 characters, 131×16 pixels CPU*
*Processor| Saturn (Lewis) Programming*
*Programming language(s)| RPN key stroke (fully merged)*
*Firmware memory| 64 KB of ROM*
*Program steps| 7200 Interfaces*
*Ports| IR (Infrared) printing Other*
*Power supply| 3×1.5V button cell batteries (Panasonic LR44, Duracell PX76A/675A or Energizer 357/303)*
*Weight| 6 oz (170 g)*
*Dimensions| 148×80×15mm*

*## Overview*

*Perhaps the HP-42S was to be released as a replacement for the aging HP-41 series as it is designed to be compatible with all programs written for the HP-41. Since it lacked expandability, and lacked any real I/O ability, both key features of the HP-41 series, it was marketed as an HP-15C replacement.*

*The 42S, however, has a much smaller form factor than the 41, and features many more built-in functions, such as a matrix editor, complex number support, an equation solver, user-defined menus, and basic graphing capabilities (the 42S can draw graphs only by programs). Additionally, it features a two-line dot matrix display, which made stack manipulation easier to understand.*

*Production of the 42S ended in 1995.[2] As this calculator is regarded amongst the best ever made in terms of quality, key stroke feel, ease of programming, and daily usability for engineers,[3] in the HP calculator community the 42S has become famous for its high prices in online auctions, up to several times its introduction price, which has created a scarcity for utility end users.*

Table 15: Duplication matches found in Textbooks by MinHash LSH deduplication (in *italics*).

| | |
|---|---|
| *# Basic Concepts in Graph Theory* | *# Basic Concepts in Graph Theory* |
| *## Section 1: What is a Graph?* | *## Section 1: What is a Graph?* |
| *There are various types of graphs, each with its own definition. Unfortunately, some people apply the term "graph" rather loosely, so you can't be sure what type of graph they're talking about unless you ask them. After you have finished this chapter, we expect you to use the terminology carefully, not loosely. To motivate the various definitions, we'll begin with some examples.* | *There are various types of graphs, each with its own definition. Unfortunately, some people apply the term "graph" rather loosely, so you can't be sure what type of graph they're talking about unless you ask them. After you have finished this chapter, we expect you to use the terminology carefully, not loosely. To motivate the various definitions, we'll begin with some examples.* |
| *Example 1 (A computer network) Computers are often linked with one another so that they can interchange information. Given a collection of computers, we would like to describe this linkage in fairly clean terms so that we can answer questions such as "How can we send a message from computer A to computer B using the fewest possible intermediate computers?"* | *Example 1 (A computer network) Computers are often linked with one another so that they can interchange information. Given a collection of computers, we would like to describe this linkage in fairly clean terms so that we can answer questions such as "How can we send a message from computer A to computer B using the fewest possible intermediate computers?"* |
| *We could do this by making a list that consists of pairs of computers that are connected. Note that these pairs are unordered since, if computer $C$ can communicate with computer $D$, then the reverse is also true. (There are sometimes exceptions to this, but they are rare and we will assume that our collection of computers does not have such an exception.) Also, note that we have implicitly assumed that the computers are distinguished from each other: It is insufficient to say that "A PC is connected to a Mac." We must specify which $PC$ and which Mac. Thus, each computer has a unique identifying label of some sort.* | *We could do this by making a list that consists of pairs of computers that are connected. Note that these pairs are unordered since, if computer $C$ can communicate with computer $D$, then the reverse is also true. (There are sometimes exceptions to this, but they are rare and we will assume that our collection of computers does not have such an exception.) Also, note that we have implicitly assumed that the computers are distinguished from each other: It is insufficient to say that "A PC is connected to a Mac." We must specify which $PC$ and which Mac. Thus, each computer has a unique identifying label of some sort.* |
| *For people who like pictures rather than lists, we can put dots on a piece of paper, one for each computer. We label each dot with a computer's identifying label and draw a curve connecting two dots if and only if the corresponding computers are connected. Note that the shape of the curve does not matter (it could be a straight line or something more complicated) because we are only interested in whether two computers are connected or not. Below are two such pictures of the same graph. Each computer has been labeled by the initials of its owner.* | *For people who like pictures rather than lists, we can put dots on a piece of paper, one for each computer. We label each dot with a computer's identifying label and draw a curve connecting two dots if and only if the corresponding computers are connected. Note that the shape of the curve does not matter (it could be a straight line or something more complicated) because we are only interested in whether two computers are connected or not. Below are two such pictures of the same graph. Each computer has been labeled by the initials of its owner.* |
| *...* | *...* |
| *## Basic Concepts in Graph Theory* | *## Basic Concepts in Graph Theory* |
| *The notation $\mathcal{P}_k(V)$ stands for the set of all $k$-element subsets of the set $V$. Based on the previous example we have* | *The notation $\mathcal{P}_k(V)$ stands for the set of all $k$-element subsets of the set $V$. Based on the previous example we have* |
| *Definition 1 (Simple graph) A simple graph $G$ is a pair $G = (V, E)$ where* | *Definition 1 (Simple graph) A simple graph $G$ is a pair $G = (V, E)$ where* |
| *- $V$ is a finite set, called the vertices of $G$, and* 
 *- $E$ is a subset of $\mathcal{P}_2(V)$ (i.e., a set $E$ of two-element subsets of $V$ ), called the edges of $G$.* | *- $V$ is a finite set, called the vertices of $G$, and* 
 *- $E$ is a subset of $\mathcal{P}_2(V)$ (i.e., a set $E$ of two-element subsets of $V$ ), called the edges of $G$.* |
| *...* | *...* |

Table 16: Near-duplication matches found in StackExchange by MinHash LSH deduplication (in *italics*).

This was originally posted on mathoverflow, but it seems it's more appropriate to post here.

*Let $B$ be a paracompact space with the property that any (topological) vector bundle $E \rightarrow B$ is trivial. What are some non-trivial examples of such spaces, and are there any interesting properties that characterize them?*

*For simple known examples we of course have contractible spaces, as well as the 3-sphere $S^3$. This one follows from the fact that its rank $n$ vector bundles are classified by $\pi_3(BO(n)) = \pi_2(O(n)) = 0$. I'm primarily interested in the case where $B$ is a closed manifold. Do we know any other such examples?*

*There is this nice answer to a MSE question which talks about using the Whitehead tower of the appropriate classifying space to determine whether a bundle is trivial or not. This seems like a nice tool (of which I am not familiar with) to approaching this problem. As a secondary question, could I ask for some insight/references to this approach?*

*EDIT Now that we know from the answer all the examples for closed 3-manifolds (integral homology spheres), I guess I can now update the question to the case of higher odd dimensions. Does there exist a higher dimensional example?*

*Let $B$ be a paracompact space with the property that any (topological) vector bundle $E \rightarrow B$ is trivial. What are some non-trivial examples of such spaces, and are there any interesting properties that characterize them?*

*For simple known examples we of course have contractible spaces, as well as the 3-sphere $S^3$. This one follows from the fact that its rank $n$ vector bundles are classified by $\pi_3(BO(n)) = \pi_2(O(n)) = 0$. I'm primarily interested in the case where $B$ is a closed manifold. Do we know any other such examples?*

*There is this nice answer to a MSE question which talks about using the Whitehead tower of the appropriate classifying space to determine whether a bundle is trivial or not. This seems like a nice tool (of which I am not familiar with) to approaching this problem. As a secondary question, could I ask for some insight/references to this approach?*

*EDIT Now that we know from the answers all the examples for closed 3-manifolds, I guess I can now update the question to the case of higher odd dimensions. Does there exist a higher dimensional example?*

---

This is a copy of my question on MSE (https://math.stackexchange.com/questions/3372432) because this forum seems better suited for historical questions:

*In 1985, Gosper used the not-yet-proven formula by Ramanujan*

$$\frac{1}{\pi} = \frac{2\sqrt{2}}{99^2} \cdot \sum_{n=0}^{\infty} \frac{(4n)!}{(n!)^4} \cdot \frac{26390n + 1103}{396^{4n}}$$

*to compute $17 \cdot 10^6$ digits of $\pi$, at that time a new world record.*

*Here (https://www.cs.princeton.edu/courses/archive/fall98/cs126/refs/pi-ref.txt) it reads:*

*There were a few interesting things about Gosper's computation. First, when he decided to use that particular formula, there was no proof that it actually converged to pi! Ramanujan never gave the math behind his work, and the Borweins had not yet been able to prove it, because there was some very heavy math that needed to be worked through. It appears that Ramanujan simply observed the equations were converging to the 1103 in the formula, and then assumed it must actually be 1103. (Ramanujan was not known for rigor in his math, or for providing any proofs or intermediate math in his formulas.) The math of the Borwein's proof was such that after he had computed 10 million digits, and verified them against a known calculation, his computation became part of the proof. Basically it was like, if you have two integers differing by less than one, then they have to be the same integer.*

*Now my historical question: Who was the first to prove this formula? Was it Gosper because he added the last piece of the proof, or was it the Borweins, afterwards? And was Gosper aware of this proof when he did his computation?*

*In 1985, Gosper used the not-yet-proven formula by Ramanujan*

$$\frac{1}{\pi} = \frac{2\sqrt{2}}{99^2} \cdot \sum_{n=0}^{\infty} \frac{(4n)!}{(n!)^4} \cdot \frac{26390n + 1103}{99^{4n}}$$

*to compute $17 \cdot 10^6$ digits of $\pi$, at that time a new world record.*

*Here (https://www.cs.princeton.edu/courses/archive/fall98/cs126/refs/pi-ref.txt) it reads:*

*There were a few interesting things about Gosper's computation. First, when he decided to use that particular formula, there was no proof that it actually converged to pi! Ramanujan never gave the math behind his work, and the Borweins had not yet been able to prove it, because there was some very heavy math that needed to be worked through. It appears that Ramanujan simply observed the equations were converging to the 1103 in the formula, and then assumed it must actually be 1103. (Ramanujan was not known for rigor in his math, or for providing any proofs or intermediate math in his formulas.) The math of the Borwein's proof was such that after he had computed 10 million digits, and verified them against a known calculation, his computation became part of the proof. Basically it was like, if you have two integers differing by less than one, then they have to be the same integer.*

*Now my historical question: Who was the first to prove this formula? Was it Gosper because he added the last piece of the proof, or was it the Borweins, afterwards? And was Gosper aware of this proof when he did his computation?*

Table 17: Exact match examples from the test set of MATH benchmark found in Textbooks by line-level exact match deduplication (in *italics*).

*Coin A is flipped three times and coin B is flipped four times. What is the probability that the number of heads obtained from flipping the two fair coins is the same?*

Video Solution

Answer:

## Problem 3.2.2 (AMC 10)

Two tour guides are leading six tourists. The guides decide to split up. Each tourist must choose one of the guides, but with the stipulation that each guide must take at least one tourist. How many different groupings of guides and tourists are possible?

......

*One morning each member of Angela's family drank an 8-ounce mixture of coffee with milk. The amounts of coffee and milk varied from cup to cup, but were never zero. Angela drank a quarter of the total amount of milk and a sixth of the total amount of coffee. How many people are in the family?*

Answer:

## Problem 20.2.15 (AMC 12)

The state income tax where Kristin lives is levied at the rate of $p\%$ of the first $28000 of annual income plus $(p+2)\%$ of any amount above $28000. Kristin noticed that the state income tax she paid amounted to $(p+0.25)\%$ of her annual income. What was her annual income?

Answer:

......

*Find the least positive integer $k$ for which the equation $\left\lfloor \frac{2002}{n} \right\rfloor = k$ has no integer solutions for $n$. (The notation $\lfloor x \rfloor$ means the greatest integer less than or equal to $x$.)*

Answer:

## Problem 40.1.9 (AIME)

Find the number of positive integers $n$ less than 1000 for which there exists a positive real number $x$ such that $n = x\lfloor x \rfloor$.', '', 'Note: $\lfloor x \rfloor$ is the greatest integer less than or equal to $x$.'

......

*What is the sum of the roots of $z^{12} = 64$ that have a positive real part?*

Answer:

## Problem 45.8.13 (AMC 12)

The complex numbers $z$ and $w$ satisfy $z^{13} = w, w^{11} = z$, and the imaginary part of $z$ is $\sin \frac{m\pi}{n}$, for relatively prime positive integers $m$ and $n$ with $m < n$. Find $n$.'

Answer:

......

Table 18: Exact match examples from the test set of MATH benchmark found in CommonCrawl by line-level exact match deduplication (in *italics*). In these examples, we only observe repeated questions from MATH, but do not identify duplicate answers.

---

*Let $x$ and $y$ be real numbers satisfying $x^4 y^5 + y^4 x^5 = 810$ and $x^3 y^6 + y^3 x^6 = 945$. Evaluate $2x^3 + (xy)^3 + 2y^3$.*

Let $x_1 < x_2 < x_3$ be the three real roots of the equation $\sqrt{2014}x^3 - 4029x^2 + 2 = 0$. Find $x_2(x_1 + x_3)$.

Let $m$ be the largest real solution to the equation

$$\frac{3}{x-3} + \frac{5}{x-5} + \frac{17}{x-17} + \frac{19}{x-19} = x^2 - 11x - 4$$

There are positive integers $a$, $b$, and $c$ such that $m = a + \sqrt{b + \sqrt{c}}$. Find $a + b + c$.

Let $f(x) = x^4 + ax^3 + bx^2 + cx + d$. If $f(-1) = -1$, $f(2) = -4$, $f(-3) = -9$, and $f(4) = -16$. Find $f(1)$.

Solve in positive integers $x^2 - 4xy + 5y^2 = 169$.

Solve in integers the question $x + y = x^2 - xy + y^2$.

Solve in integers $\frac{x+y}{x^2 - xy + y^2} = \frac{3}{7}$

Prove the product of $4$ consecutive positive integers is a perfect square minus $1$.

For any arithmetic sequence whose terms are all positive integers, show that if one term is a perfect square, this sequence must have infinite number of terms which are perfect squares.

Prove there exist infinite number of positive integer $a$ such that for any positive integer $n$, $n^4 + a$ is not a prime number.

......

---

*The real root of the equation $8x^3 - 3x^2 - 3x - 1 = 0$ can be written in the form $\frac{\sqrt[3]{a} + \sqrt[3]{b} + 1}{c}$, where $a$, $b$, and $c$ are posit ive integers. Find $a + b + c$.*

Find the number of positive integers $m$ for which there exist nonnegative integers $x_0$, $x_1$, ... , $x_{2011}$ such that

$$m^{x_0} = \sum_{k=1}^{2011} m^{x_k}.$$

Suppose $x$ is in the interval $[0, \frac{\pi}{2}]$ and $\log_{24\sin x}(24\cos x) = \frac{3}{2}$. Find $24\cot^2 x$.

Let $P(x)$ be a quadratic polynomial with real coefficients satisfying $x^2 - 2x + 2 \le P(x) \le 2x^2 - 4x + 3$ for all real numbers $x$, and suppose $P(11) = 181$. Find $P(16)$.

Let $(a, b, c)$ be the real solution of the system of equations $x^3 - xyz = 2$, $y^3 - xyz = 6$, $z^3 - xyz = 20$. The greatest possible value of $a^3 + b^3 + c^3$ can be written in the form $\frac{m}{n}$, where $m$ and $n$ are relatively prime positive integers. Find $m + n$.

Find the smallest positive integer $n$ with the property that the polynomial $x^4 - nx + 63$ can be written as a product of two nonconstant polynomials with integer coefficients.

The zeros of the function $f(x) = x^2 - ax + 2a$ are integers. What is the sum of the possible values of $a$?

Let $a$, $b$, and $c$ be three distinct one-digit numbers. What is the maximum value of the sum of the roots of the equation $(x - a)(x - b) + (x - b)(x - c) = 0$?

At the theater children get in for half price. The price for 5 adult tickets and 4 child tickets is 24.50. How much would 8 adult tickets and 6 child tickets cost?

The quadratic equation $x^2 + px + 2p = 0$ has solutions $x = a$ and $x = b$. If the quadratic equation $x^2 + cx + d = 0$ has solutions $x = a + 2$ and $x = b + 2$, what is the value of d?

......

---

*Find the smallest positive integer $n$ with the property that the polynomial $x^4 - nx + 63$ can be written as a product of two nonconstant polynomials with integer coefficients.*

The zeros of the function $f(x) = x^2 - ax + 2a$ are integers. What is the sum of the possible values of $a$?

Let $a$, $b$, and $c$ be three distinct one-digit numbers. What is the maximum value of the sum of the roots of the equation $(x - a)(x - b) + (x - b)(x - c) = 0$ ?

At the theater children get in for half price. The price for 5 adult tickets and 4 child tickets is 24.50. How much would 8 adult tickets and 6 child tickets cost?

The quadratic equation $x^2 + px + 2p = 0$ has solutions $x = a$ and $x = b$. If the quadratic equation $x^2 + cx + d = 0$ has solutions $x = a + 2$ and $x = b + 2$, what is the value of d?

PolynomialAndEquation Root Delta SpecialEquation Function NumberTheoryBasic IndeterminateEquation SqueezeMethod Pythagore anTripletFormula TrigIdentity Inequality LogicalAndReasoning

AMC10/12 AIME IMO

US International

With Solutions

© 2009 - 2023 Math All Star

......

Table 19: Exact match examples from the test set of MATH benchmark (upper) and MMLU-STEM (bottom) found in OpenWebMath by line-level exact match deduplication (in *italics*). In these examples, we only observe repeated questions, but do not identify duplicate answers.

*The sum of an infinite geometric series is a positive number S, and the second term in the series is 1. What is the smallest possible value of S?*

**(A)** $\frac{1+\sqrt{5}}{2}$    **(B)** 2    **(C)** $\sqrt{5}$    **(D)** 3    **(E)** 4

## Problem 17

All the numbers $2, 3, 4, 5, 6, 7$ are assigned to the six faces of a cube, one number to each face. For each of the eight vertices of the cube, a product of three numbers is computed, where the three numbers are the numbers assigned to the three faces that include that vertex. What is the greatest possible value of the sum of these eight products?

**(A)** 312    **(B)** 343    **(C)** 625    **(D)** 729    **(E)** 1680

...

*What is the value of $b + c$ if $x^2 + bx + c > 0$ only when $x \in (-\infty, -2) \cup (3, \infty)$?*

May 11, 2020

...

*An ambulance travels at 40 mph and can follow a 20-mile route making no stops to get to the hospital. A helicopter travels at one mile per minute, and the air route is 15 miles to get to the same hospital. However, the helicopter takes three minutes for takeoff and three minutes for landing. How many fewer minutes does it take for the helicopter to complete its trip (takeoff, flight and landing) than for the ambulance to complete its trip?*

Apr 6, 2020

#1
+34
0

Keep in mind that Time=Distance/Speed

---

*What is the greatest possible area of a triangular region with one vertex at the center of a circle of radius 1 and the other two vertices on the circle?*

A bad first step is to put the center at the origin, one point at (1,0) , and one point at (sin x, cos x).

A start is the area of a triangle with included angle expression

$$\frac{a \times b \times \sin \theta}{2}$$

Assuming $\theta$ in radians. If theta is $\pi/2$ then we have a right triangle. Let a=b=1. Area expression is

$$A = (\sin \theta)/2$$

This is maximum for $\theta = \pi/2$.

Answer is maximum area for a right triangle.

...

---

## G  Evaluation of Continal Pre-trained Models on General Langauge Benchmarks

Does continual pre-training on MATHPILE lead to improvements in general language benchmarks? To explore this, we evaluated some continual pre-trained models on MATHPILE (in Table 4) in several representative benchmarks including PIQA [8], ARC-Easy [15], ARC-Challenge [15], SciQ [64] and HellaSwag [69]. For these evaluations, we used the infrastructure provided by EleutherAI's lm-evaluation-harness, [7] and following common practices, we report the accuracy (acc norm) metric.

Table 20: Performance of continual pre-trained models on general language benchmarks

| Models | PiQA | ARC-Challenge | ARC Easy | SciQ | Hallswag |
|---|---|---|---|---|---|
| Mistral-7B | 81.93 | 53.75 | 79.58 | 94.0 | 81.05 |
| + Textbooks | 80.14 | 52.73 | 79.92 | 95.6 | 81.15 |
| + Wikipedia | 80.57 | 56.48 | 79.71 | 94.8 | 82.07 |
| + Stackexchange | 80.41 | 49.66 | 75.38 | 90.5 | 82.87 |

One important point that needs to be emphasised is that when enhancing a model's capabilities in a specific domain, it is typically necessary to mix the new domain-specific training data with the original training data, which helps to prevent catastrophic forgetting. In current experiments, we conducted continual pre-training exclusively on math-specific data, which means that we did not necessarily expect improvements in general language modeling benchmarks and, in some cases, a regression could occur. As shown in Table 20, after continual pre-training on MATHPILE subsets, the model's general language modeling abilities did not experience significant degradation. In fact, there were some improvements on certain benchmarks, though some metrics did see slight declines.

---

[7]https://github.com/EleutherAI/lm-evaluation-harness

