# OpenReview forum: "MathPile: A Billion-Token-Scale Pretraining Corpus for Math"
_NeurIPS.cc/2024/Datasets_and_Benchmarks_Track — NeurIPS 2024 Track Datasets and Benchmarks Poster_

### Official Review · Reviewer_t4e7 · 2024-07-23
**High quality and diverse dataset for mathematical reasoning**

**Rating:** 9
**Confidence:** 4
**Correctness:** The dataset is constructed in a sound…
**Clarity:** The paper is well-written and easy to…

**Review:**

The presented dataset is comprehensive and high-quality. The work is described in detail and with clarity. The significance of this work is high, benefiting applications in education tools. The open-sourced dataset demo and materials (with Github, project page) are clear and comprehensive.

**Strengths:**

Strengths
* Dataset is large-scale and collected from diverse Math sources
* The data processing pipeline seems reasonable
* The open-sourced dataset, as well as its demos, project pages, and documentation has high quality

**Additional Feedback:**

None.

**Documentation:**

The documentation of the work is of high quality.

**Ethics:**

No ethical concerns.

**Limitations:**

Limitations
* Construct a multilingual mathematical reasoning dataset.

**Opportunities For Improvement:**

* It would be great to include some concrete dataset examples from different sources at different levels in Sec. 4.1. Statistics.

**Relation To Prior Work:**

This work clearly differentiates itself from prior work regarding its math focus, diversity of sources, and commitment to producing high-quality data and data documentation.

**Summary And Contributions:**

This work contributes a large-scale, high-quality, and diverse dataset for mathematical reasoning. The dataset is constructed from a wide range of math sources using a chain of reasonable methods. In addition, the data quality is assessed via data contamination detection. The dataset is presented and analyzed with detailed statistics and model pretraining experiments.

---

> ### Author Rebuttal · Authors · 2024-08-17
>
> Thank you very much for your recognition and positive feedback! We sincerely appreciate your high rating and are delighted that our work resonated with you.
>
> > It would be great to include some concrete dataset examples from different sources at different levels in Sec. 4.1. Statistics.
>
> Thank you for the helpful suggestions. We have actually included several examples from different sources in `Appendix C Examples of MathPile` (shown in Figures 4–10). We apologize if these were overlooked, and we are more than happy to add a wider variety of examples to further illustrate our MathPile dataset.
>
> > Limitations: Construct a multilingual mathematical reasoning dataset.
>
> Thank you for your insightful suggestion. We fully agree that extending mathematical reasoning to multiple languages is both important and urgent. This is indeed a key direction for our future work.

---

> > ### Comment · Reviewer_t4e7 · 2024-08-30
> > **Thank you for authors' feedback**
> >
> > Thank you for pointing out the examples in `Appendix C Examples of MathPile`.
> > Since I do not have many requests for change, I maintain my score

---

### Official Review · Reviewer_y6yr · 2024-07-23
**Billion-Token-Scale Pretraining Corpus for Math - Strong Contribution but Limited Results**

**Rating:** 6
**Confidence:** 4
**Correctness:** Yes, the claims made in the paper are…
**Clarity:** Yes paper is well written.

**Review:**

This paper makes a significant contribution by releasing a large-scale corpus to the open-source community to enhance the math performance of large language models (LLMs). The data preprocessing pipeline, including data deduplication, contamination detection, and both local and global preprocessing, is a valuable contribution in itself, as it will benefit those curating pre-training corpora for math and other domains. Additionally, the paper is well-written and clearly presented. However, it has limited experiments, using only one base model for pre-training, lacks strong baselines, and shows limited performance improvement as more data is added. I have highlighted the strengths and limitations of the work in more detail below.

**Strengths:**

1. **High-Quality and Diverse Corpus**: MathPile is a large-scale pre-training corpus containing approximately 9.5 billion tokens, specifically designed to enhance the math performance of large language models (LLMs).

2. **Rigorous Data Processing Pipeline**: The authors have implemented a comprehensive and math-specific data processing pipeline that includes preprocessing, prefiltering, language identification, cleaning and filtering, and deduplication. This ensures the high quality and reliability of the corpus. Moreover, Table 5 shows ablations for the data pre-processing part that highlights the pipeline’s effectiveness.

3. **Data Contamination Detection**: The study includes a robust data contamination detection mechanism to eliminate duplicates from widely used mathematical reasoning benchmark test sets, ensuring the integrity and uniqueness of the dataset.

4. **Open-Source Contribution**: By releasing MathPile to the open-source community, the paper provides a valuable resource that can be used to improve the performance of LLMs in the math domain, benefiting researchers and practitioners.

5. **Well-Written and Clearly Presented**: The paper is well-structured and clearly presented, making the methodologies and findings accessible and understandable to the research community.

**Additional Feedback:**

I have mentioned the strengths and limitations of the work in the review above.

**Documentation:**

Yes, the availability, maintenance, and ethical and responsible use is specified in the paper.

**Ethics:**

No.

**Limitations:**

Yes.

**Opportunities For Improvement:**

**Experiments with More Base Models**

The paper only includes pre-training with the Mistral 7B base model. However, conducting more experiments on stronger base models from the LLama or Qwen families, or more advanced math models like DeepSeek 7B, would better demonstrate the effectiveness of the proposed corpus.

**Limited Improvements**

The second-to-last row in Table 4 shows very limited or no improvements when more data is added from various sources. In comparison, row 2, which uses only 0.56B tokens of textbook data, shows some gains. This raises questions about the effectiveness of the entire pre-training corpus. Additionally, there are no experiments with the full pre-training corpus. Was this due to a lack of time, or were there no improvements when the full corpus was used? Moreover, there are not significant improvements in GSM8K, MATH, and MMLU-MATH across the board in Table 4.

**Performance on Non-Math Benchmarks**

Although the paper includes a good number of math benchmarks, it should also show performance on other general tasks in the Open-LLM leaderboard to ensure there are no significant regressions in performance on non-math benchmarks.

**Performance Comparison with Baselines in Table 1**

Table 4 only compares performance with one pre-training corpus, OpenWebMath. However, Table 1 highlights the strengths and weaknesses of various math pre-training corpora, but there is no comparison with those datasets.

**Miscellaneous Comments**

- Most benchmarks evaluate models based on the final answer. It would be helpful to see how the reasoning improves specifically. For instance, showing how reasoning steps improve when models are fine-tuned on MathPile, possibly through anecdotes or human evaluation on a subset of data, would be beneficial.
- How are false positives in data deduplication detected?
- In line 108, how is recall measured?
- What was the design choice behind this specific method in data contamination detection? Does any other work use a similar method?

**Relation To Prior Work:**

Yes, the related work has been discussed in detail.

**Summary And Contributions:**

The paper introduces MathPile, a high-quality and diverse pre-training corpus specifically designed for the math domain. This extensive corpus contains approximately 9.5 billion tokens. The authors have carefully sourced and compiled the data, implementing a rigorous and math-specific pipeline. This pipeline includes several stages: preprocessing, prefiltering, language identification, cleaning and filtering, and deduplication, all aimed at ensuring the corpus's high quality. Additionally, the study performs data contamination detection to eliminate duplicates from widely used mathematical reasoning benchmark test sets.

---

> ### Author Rebuttal · Authors · 2024-08-30
>
> > Performance Comparison with Baselines in Table 1: Table 4 only compares performance with one pre-training corpus, OpenWebMath. However, Table 1 highlights the strengths and weaknesses of various math pre-training corpora, but there is no comparison with those datasets.
>
> Thank you very much for pointing this out. We primarily focused on comparing with OpenWebMath because it is the most recent and closely related work, and the comparison was conducted through direct pre-training. To address your concern, we selected two open-source corpora listed in Table 1, `DM-Mathematics` and `AMPS`, and conducted pre-training comparisons with them.
> The experimental results are as follows:
>
> | **Models**             | **GSM8k** | **MATH** | **SAT-MATH** | **MMLU-MATH** | **MathQA** | **AQuA** |
> |------------------------|:---------:|:--------:|:------------:|:-------------:|:----------:|:--------:|
> | Mistral-7B-v0.1             |   47.38   |   10.08  |     47.27    |     44.92     |    23.51   |   27.95  |
> | w/ DM-Mathematics (5B) |     0     |     0    |       0      |       0       |      0     |     0    |
> | w/ AMPS (1B)           |    0.08   |   0.82   |     3.18     |      0.47     |    10.99   |   8.27   |
>
> Note that each dataset was trained for at least one epoch, given the constraints of limited computational resources.
>
> We found that directly pre-training with these datasets actually degraded the model's performance, rather than improving it. We believe this phenomenon occurs because both DM-Mathematics and AMPS only consist of problem sets that are relatively short and monotonous in their presentation, making them unsuitable for direct pre-training. Typically, such datasets are better suited for pre-training when mixed with more general corpora (refer to Lemur [1]). This process involves many decisions and is further complicated by factors such as dataset size, making it challenging to compare these problem-set datasets with our MathPile corpus in a controlled setting.
>
> We hope these additional experiments help alleviate some of your concerns.
>
>
> > Performance on Non-Math Benchmarks: Although the paper includes a good number of math benchmarks, it should also show performance on other general tasks in the Open-LLM leaderboard to ensure there are no significant regressions in performance on non-math benchmarks.
>
> Thank you for your valuable feedback. In this paper, our goal was to demonstrate the specific improvements in math-related tasks, which are the primary target of this corpus.  We acknowledge that evaluating the model on general tasks is essential to ensure that there are no significant regressions in performance on non-math benchmarks.
>
> One important point we'd like to emphasize—and hope we can agree on—is that when enhancing a model's capabilities in a specific domain, it is typically necessary to mix the new domain-specific training data with the original training data, which helps to prevent catastrophic forgetting [1, 2,3,4]. In our current experiments, we conducted continual pre-training exclusively on math-specific data, which means that we did not necessarily expect improvements in general language modeling benchmarks and, in some cases, regression could occur.
>
> To address your concerns, we conducted additional tests on several common benchmarks to evaluate the general language modeling capabilities of our model after pre-training with different subsets of MathPile. The results are as follows:
>
> |     **Models**    |  **PiQA** | **ARC Challenge** | **ARC Easy** | **SciQ** | **Hallswag** |
> |:-----------------:|:---------:|:-----------------:|:------------:|:--------:|:------------:|
> |     Mistral-7B    | **81.93** |       53.75       |     79.58    |   94.0   |     81.05    |
> |    w/ textbooks   |   80.14   |       52.73       |   **79.92**  | **95.6** |     81.15    |
> |   w/ Wikipedia    |   80.57   |     **56.48**     |     79.71    |   94.8   |     82.07    |
> | w/ Stackexchange  |   80.41   |       49.66       |     75.38    |   90.5   |   **82.87**  |
>
> As we can see, after the additional pre-training, the model's general language modeling abilities did not experience significant degradation. In fact, there were some improvements on certain benchmarks, though some metrics did see slight declines. We hope that these experimental results help alleviate your concerns.
>
> Note: For these evaluations, we used the infrastructure provided by EleutherAI’s lm-evaluation-harness (https://github.com/EleutherAI/lm-evaluation-harness), and following common practices, we report the accuracy (`acc norm`) metric.
>
>
> References:
>
> 1. Lemur: Harmonizing Natural Language and Code for Language Agents. ICLR 2024.
>
> 2. Continual Pre-Training of Large Language Models: How to (re)warm your model? ICML workshop 2023
>
> 3. SeaLLMs - Large Language Models for Southeast Asia. ACL 2024
>
> 4. Typhoon: Thai Large Language Models. 2023

---

> ### Author Rebuttal · Authors · 2024-08-30
>
> Thanks for your valuable feedback and suggestions. Sorry for the late response due to conducting experiments and other commitments.
>
> > Limited Improvements: The second-to-last row in Table 4 shows very limited or no improvements when more data is added from various sources. In comparison, row 2, which uses only 0.56B tokens of textbook data, shows some gains. This raises questions about the effectiveness of the entire pre-training corpus.
>
> The observed limited improvements when incorporating data from various sources, as highlighted in Table 4, can be attributed to the varying quality and distribution of the datasets. Specifically, the data quality across different sources is not uniform, and the distribution of data from each source does not necessarily align well with the specific mathematical reasoning tasks measured by benchmarks like GSM8K, MATH, and MMLU-MATH. For example, textbook data, although high in quality, is inherently scarce. On the other hand, the richness of web page data, however, represents a significant potential resource for expanding the math-centric corpus, and we view this as a key focus for our future work. By enhancing our collection of mathematical web pages, we aim to increase the volume of relevant data, which could lead to more significant improvements in downstream performance.
>
> > Was this due to a lack of time, or were there no improvements when the full corpus was used?
>
> As you correctly noted, experiments with the full pre-training corpus were not included. This was primarily due to limitations in computational resources, which constrained our ability to conduct exhaustive experiments with the full corpus. However, we have conducted preliminary validation by sampling from each subset (if it is large enough), and we hope this can be understood and appreciated. We also acknowledge that data mixing is indeed a worthwhile area for further exploration, and we plan to conduct more comprehensive data mixing experiments in our future work.

---

### Official Review · Reviewer_yX5o · 2024-07-30
**A good data  for Math**

**Rating:** 6
**Confidence:** 4
**Correctness:** good
**Clarity:** good

**Review:**

Pros:
- Diversity: This work devotes significant effort to gathering data from a variety of sources, which is labor-intensive but highly valuable to the community.
- High-Quality: This work develops heuristic rules for both per-subset and global data processing to enhance data quality, conducting an ablation study for each subset.
- Data Contamination: This work conducts data contamination detection to remove duplicates from downstream mathematical benchmark test sets.

Cons:
- Quality Definition: Throughout the pipeline, the quality of this work is largely determined by heuristic rules that focus mainly on parsing, cleaning, and filtering. However, it lacks a formal definition of quality to guide the pipeline effectively.
- Distance with Downstream Test Sets: Compared with OpenWebMath in terms of data contamination, this work shows interesting results, where questions from MATH and MMLU-STEM hit OpenWebMath much more frequently than this work (e.g., 195 on MATH of OpenWebMath vs 23 on MATH of this work). This suggests that OpenWebMath is closer to the downstream test sets, which may potentially lead to better performance.
- Data Mixing: This seems to be a significant problem for users intending to use this dataset since the combined mixture (Textbooks, Wikipedia, StackEx, CC) performs worse than Textbooks only, as shown in Table 4. This could hinder users from directly using this dataset to explore further enhancements in mathematical reasoning.

**Strengths:**

Please see Pros.

**Additional Feedback:**

see the cons

**Documentation:**

good

**Limitations:**

Please see Cons.

**Opportunities For Improvement:**

Explain the limitations of this work along the same axes as above.

**Relation To Prior Work:**

good

**Summary And Contributions:**

This paper presents a diverse and high-quality, math-centric corpus. The collection of the corpora includes mathematical textbooks, ArXiv papers, Wikipedia, ProofWiki, StackExchange, and web pages from the Common Crawl. It also conducts global data processing, including language identification, data cleaning and filtering, data deduplication, and data contamination detection. The resulting corpora consist of 9.5B tokens. The paper conducts continual pre-training experiments to explore the effectiveness of each subset, showing an average improvement of up to 5%. It open-sources the code and data to the community.

---

> ### Author Rebuttal · Authors · 2024-08-30
>
> Thanks for your insightful feedback. Sorry for the late response due to conducting experiments and other commitments.
>
> > Quality Definition: Throughout the pipeline, the quality of this work is largely determined by heuristic rules that focus mainly on parsing, cleaning, and filtering. However, it lacks a formal definition of quality to guide the pipeline effectively
>
> Thank you for your insightful comment. Defining the quality of pre-training corpora is indeed a critical issue, but it remains an open and challenging problem, especially in the era of large-scale pre-training for language models. A well-established and scientific framework for defining data quality, along with a reliable evaluation model, would indeed be appealing. Such a framework could guide us in iterating on cleaning rules based on clear quality indicators. However, this is quite difficult to achieve, especially for a domain-specific dataset like ours in the math field, which involves unique characteristics such as mathematical symbols and specialized expressions.
>
> Common approaches, like using a reference model to measure perplexity or score the cleaned corpus, may not be reliable or scalable, particularly when using large models. More importantly, these scores may not directly correlate with the specific mathematical reasoning capabilities that we aim to enhance through pre-training.
>
> For these reasons, we adopted the most direct approach: pre-training on the corpus and evaluating its performance on mathematical reasoning benchmarks to reflect data quality. This approach is also widely used in practice. Additionally, we conducted ablation studies on different stages of our data cleaning process (using Wikipedia as an example, cf. `Sec. 4.2 The Effectiveness of Data Processing Pipeline`) to validate the effectiveness of our methods.
>
> We hope this explanation clarifies our approach and that we can reach a mutual understanding on this point.
>
> > Distance with Downstream Test Sets: Compared with OpenWebMath in terms of data contamination, this work shows interesting results, where questions from MATH and MMLU-STEM hit OpenWebMath much more frequently than this work (e.g., 195 on MATH of OpenWebMath vs 23 on MATH of this work). This suggests that OpenWebMath is closer to the downstream test sets, which may potentially lead to better performance.
>
> Thank you for your insightful observation. You are absolutely correct that during our data contamination analysis, we found that OpenWebMath contains more samples overlapping with downstream test sets like MATH and MMLU-STEM. To eliminate the impact of data leakage, we carefully removed these leaked samples from both MathPile and OpenWebMath before conducting pre-training experiments for comparison. We hope this clarification helps to align our understanding of the paper and avoids any potential misunderstandings.
>
> > Data Mixing: This seems to be a significant problem for users intending to use this dataset since the combined mixture (Textbooks, Wikipedia, StackEx, CC) performs worse than Textbooks only, as shown in Table 4. This could hinder users from directly using this dataset to explore further enhancements in mathematical reasoning.
>
> Thank you for bringing up this important point. While it is true that the combined mixture of Textbooks, Wikipedia, StackExchange, and Common Crawl generally performs worse than using only the Textbooks subset, there are specific cases, such as the GSM8K benchmark, where the mixed data actually outperforms Textbooks alone. This highlights the inherent challenge of data mixing, especially when combining sources with different data qualities and distributions. Differences in format, content focus, and distribution across sources make it difficult to balance performance across multiple benchmarks during training. We fully acknowledge this issue and have mentioned it in our paper, noting that “We believe further exploration of data mixing is necessary.” We are currently conducting additional experiments to refine the data mixing strategy. However, since these experiments are ongoing, we cannot yet provide definitive conclusions.
>
> It is also important to emphasize that MathPile does not need to be used as a whole. Users can select specific subsets based on their particular needs. For instance, recent work like JiuZhang 3.0 [1] has already leveraged some subsets of MathPile (e.g., Textbooks, arXiv, and Wikipedia) to synthesize training data. Additionally, a few subsets of MathPile have been utilized for benchmarking KNN-LMs [2], beyond the common use cases like pre-training. We believe that allowing users to choose subsets that align with their specific objectives is one of MathPile's key strengths.
>
> While data mixture remains a challenging problem that we are actively exploring, we hope this explanation clarifies our work and offers additional context on how MathPile could be effectively utilized in various scenarios, depending on users’ specific objectives and needs.
>
> References:
>
> [1]. Zhou et al., JiuZhang3.0: Efficiently Improving Mathematical Reasoning by Training Small Data Synthesis Models.  2024
>
> [2]. Geng et al., Great Memory, Shallow Reasoning: Limits of kNN-LMs. 2024

---

### Official Review · Reviewer_W3kS · 2024-08-01
**A large scale dataset for mathematics with some issues**

**Rating:** 6
**Confidence:** 4
**Correctness:** (See "Review" section)
**Clarity:** (See "Review" section)

**Review:**

The paper presents an important addition to the field. The datasource are clearly described. I appreciated that a language model (Mistral 7B) was pretrained to test the effectiveness of this dataset in terms of how much the scores improve.

Nonetheless, there are a few issue that I found striking. I am missing a section on which parts of the data were cleaned by scripts and which by humans (if at all). Also, the script are missing - it seems impossible to reproduce the dataset. A section should be added to clearly explain this, as well as to add code.

Therefore, it looks like a very promising dataset, but a number of issues and missing best-practices make it slightly problematic.

**Strengths:**

(see Review section)

**Additional Feedback:**

How did you handle the copyright regarding textbooks in your dataset?

Why have you not made the cleaning scripts available? In your GitHub repo you state "The decisions made during the data collection and processing phases might not always be optimal." It is hard to follow these, and assess how suboptimal they were if the scripts are not made public.

From section 4.2 it seems to me that the quality of filtering was solely assessed by pertaining Mistral's model. Is that correct?

Could you explain how it could happen that the score of Mistral7B in table 5 is slightly higher on MATH in the default version of the language models and then drops when pretrained on the cleaned Wikipedia (while performance on all other benchmarks rises)?
This seems strange to me - and any explanations should also be included in the paper

**Documentation:**

(See "Review" section)

**Limitations:**

lines 371: " for example, many detailed decisions in its creation were made empirically, which may not always be optimal, and verifying decisions directly can be difficult"

It is not clear from the text what these were, as some limitation explanations are scattered throughout the text. Could you make a section where all limitations are explained?

**Opportunities For Improvement:**

some typos: line 18-19 "our concern centers on mathematical reasoning [...] for which can boost"

**Relation To Prior Work:**

(See "Review" section)

**Summary And Contributions:**

A large-scale mathematical dataset is introduced, suitable for model pre-training, that goes beyond web-page content and includes a wide array of sources, and aims to be well documented. Detailed comparisons with existing corpora are made, statistics on documents lengths from various sources are given, and a pre-training experiment is carried out.

---

> ### Author Rebuttal · Authors · 2024-08-17
>
> > How did you handle the copyright regarding textbooks in your dataset?
>
> Thank you for raising this important question. We have been extremely cautious in addressing copyright concerns while collecting data, and we have included a detailed discussion on this matter in `Appendix B Ethics Statement`, which you are welcome to review. Specifically, regarding the potential copyright issues related to textbooks, we followed these guidelines:
>
> **1. Strict Selection of Openly Licensed Materials**: We only included textbooks that are distributed under open licenses, such as CC BY licenses, and are freely available for download. We carefully excluded any textbooks that are prohibited from redistribution.
>
> **2. Adherence to Fair Use Principles**: Our dataset is intended solely for research purposes and not for commercial use. We believe that using this corpus for scientific research falls under the fair use doctrine, significantly reducing the potential for any negative impact on the rights holders.
>
> **3. Responsiveness to Feedback**: We acknowledge that it is challenging to obtain explicit permission from every author for every textbook included in the dataset. Therefore, we are fully open to receiving feedback from users and textbook authors. If any author requests the removal of their content, we will strictly adhere to those requests and promptly take action.
>
> We believe that by strictly implementing these measures, we can minimize copyright issues to the greatest extent possible, and we hope for your understanding.
>
>
> > From section 4.2 it seems to me that the quality of filtering was solely assessed by pertaining Mistral's model. Is that correct?
>
> Indeed, the quality of the cleaned data is primarily assessed through the pre-training of the Mistral model and the subsequent performance on mathematical reasoning benchmarks, as indicated in `Section 4.2`. We believe this approach is intuitive since the main purpose of collecting mathematical corpora is to enhance the mathematical reasoning capabilities of existing models. Additionally, we provide some statistical information in `Section 4.1` to further reflect the quality of the dataset to some extent. We also conducted ablation studies, as detailed in `The Effectiveness of Data Processing Pipeline` section, to validate the impact of our cleaning process.
>
> While we acknowledge that there may be other methods to evaluate corpus quality—whether statistical or model-based—resulting in a quality score, such scores may not always directly correlate with the intended improvement in mathematical reasoning. We believe that demonstrating data quality through pre-training and performance on relevant benchmarks is the most straightforward and reliable approach, and we hope we can reach a consensus on this point.
>
>
> > Could you explain how it could happen that the score of Mistral7B in table 5 is slightly higher on MATH in the default version of the language models and then drops when pretrained on the cleaned Wikipedia (while performance on all other benchmarks rises)? This seems strange to me - and any explanations should also be included in the paper
>
> We appreciate your astute observation regarding the slight drop in MATH benchmark performance after pre-training on cleaned Wikipedia data, while other benchmarks showed improvement. This phenomenon indeed warrants further discussion. We believe this observation can be attributed to two main factors:
>
> **1. Statistical Fluctuations**: The difference in MATH performance (10.08 vs. 9.96) is notably small, about 0.12 percentage points. Such minor variations are often within the range of statistical fluctuations inherent in the training process of large language models. Given the complexity of these models and the stochastic nature of training, small performance variations across different runs or datasets are not uncommon. To address this, we commit to conducting additional experimental runs in our revised paper to verify the consistency of this phenomenon and provide a more robust statistical analysis.
>
> **2. Training Dynamics**: During the pre-training process, models often undergo complex learning dynamics where improvements in certain abilities might temporarily come at the cost of others. This is particularly true when adapting to a new domain or dataset. The model may be in a transitional state where it's adapting to the Wikipedia-style mathematical content, which could temporarily affect its performance on the MATH benchmark. It's possible that with extended training or a larger dataset, this temporary trade-off would be resolved.
>
> In our revised paper, we will include a more detailed analysis of these factors, including results from additional experimental runs. We thank the reviewer for highlighting this interesting observation, as it provides valuable insights into the nuanced effects of pre-training on specific mathematical reasoning capabilities.

---

> > ### Comment · Reviewer_W3kS · 2024-09-01
> >
> > I thank the authors for their comprehensive feedback and for following up on my remarks. Nonetheless, I feel some of the issues still need work to be fully resolved:
> >
> > 1) All in all, since this is a very complex dataset made up of many sources of data (which themselves are complex), I feel it is *absolutely necessary* to have a better understanding of how the dataset was created. First, a diagram should be given where it is made clear which steps were automatic and which were human, with pointers to the various sections where more explanations are given. Then, there should be a clearer separation between which parts were done automatically and which by hand. The scripts (thank you for pointing out the link!) should be accompanied by a high-level description of what data they ingest and what data they output.
> >
> > 2) Regarding reproducibility, since this can probably not be achieved in principle because of human dataset curation (which is fine). This makes it all the more important to list *all* steps in which humans intervened.
> > Some of your explanations are slightly vague; for example, when you mention above in rebuttal: "creating the dataset involved making numerous decisions, including but not limited to the following:" It would be great if this list could be comprehensive. I realize that it may be some effort to create this post-dataset release, but I believe it would help a lot to gain a better understanding of the dataset.
> > Similarly, when you mention in the table "using FastText with custom thresholds" or "automated application of rules for content filtering and removal", questions arise: what thresholds? Which kinds of rules were used? How many rules? (That an automated application of rules was carried out and was already kind of clear.)
> >
> > 3) I would encourage the authors to make a more detailed set of experiments, on the one hand, to get a more rigorous result regarding the quality of the filtering and, on the other hand, to see whether the MATH score was an artefact or not.
> >
> > I will keep my score, but I am certain that if the authors invest some more time in this dataset, it will make a great contribution. If it gets accepted now, this should be done for the camera-ready version; if it doesn't get accepted, I recommend the authors to improve these issues and resubmit at their earliest convenience to another flagship venue.

---

> > > ### Author Response · Authors · 2024-09-01
> > > **Rebuttal By Authors**
> > >
> > > We sincerely thank you for your detailed feedback. We agree with your suggestion to more explicitly indicate which parts of the dataset creation process were automated and which required human decision-making in paper and scripts. We will include a detailed introduction about it in our revision. Additionally, we would like to clarify the following points:
> > >
> > > **Regarding Dataset Reproducibility**: We appreciate your concerns regarding reproducibility and believe there might be a slight misunderstanding regarding what is achievable with our dataset. It is true that human judgment played a role in initial decisions, such as setting thresholds and designing rules. However, once these decisions were made, they were codified into our scripts, which are openly available and executed automatically.
> > >
> > > About the question "what thresholds? Which kinds of rules were used? How many rules?" For instance, the thresholds include those used with FastText to filter non-English documents. Actually, the specifics of these rules and thresholds are thoroughly detailed in `Appendix E Details of Global Data Processing` of our paper, and their implementations are accessible in our GitHub repository (https://github.com/GAIR-NLP/MathPile/tree/main/src), facilitating easy reproduction of the dataset by running these scripts.
> > >
> > > **Beyond Reproducibility**:  Moreover, we also would like to emphasize that data has become akin to fuel in the era of large language models, driving advancements in deep learning—similar to landmark datasets like ImageNet. Not everyone has the resources to replicate complete dataset creation processes from scratch. Instead, many researchers focus on how to effectively utilize these datasets. Our dataset is fully reproducible, as detailed in our paper and demonstrated by the code available in our GitHub repository. We hope to reach a consensus on this point and for you to recognize the significant contribution our open-sourced dataset makes to the field. Already, many researchers are using our dataset for model training [1], data synthesis [2], and benchmarking [3], which underscores its value and impact.
> > >
> > > **On the Need for Detailed Experiments**: We understand and value the need for detailed experiments to verify the quality of data filtering. However, in the current era of large language models, the scale of data and computational requirements has grown exponentially, making extensive experimentation at every step of dataset creation both challenging and costly. This shift necessitates a change in approach to evaluating scientific contributions in our field. Nonetheless, we have endeavoured to demonstrate the effectiveness of our data-cleaning pipeline through targeted experiments discussed in `Section 4.2 The Effectiveness of Data Processing Pipeline`.
> > >
> > > We are grateful for your insights and hope that our responses can dispel your concerns. We truly appreciate the effort and time you have taken to review our work and hope you understand the challenges and efforts involved in creating this dataset, as well as its necessity for advancing this field.
> > >
> > > References:
> > >
> > > 1. BAM! Just Like That: Simple and Efficient Parameter Upcycling for Mixture of Experts. 2024
> > >
> > > 2. JiuZhang3.0: Efficiently Improving Mathematical Reasoning by Training Small Data Synthesis Models. 2024
> > >
> > > 3. Great Memory, Shallow Reasoning: Limits of kNN-LMs. 2024

---

> ### Author Rebuttal · Authors · 2024-08-17
>
> > lines 371: " for example, many detailed decisions in its creation were made empirically, which may not always be optimal, and verifying decisions directly can be difficult" It is not clear from the text what these were, as some limitation explanations are scattered throughout the text. Could you make a section where all limitations are explained?
>
> Thank you for your insightful suggestion. Indeed, creating the dataset involved making numerous decisions, including but not limited to the following:
> - **Data Source Selection**: Choices such as selecting specific common crawl dumps or determining which subject areas of arXiv papers to include.
> - **(Pre)Filtering Thresholds**: Decisions like filtering out StackExchange questions and answers based on community scores, or setting the thresholds for language identification using fastText.
> - **Data Cleaning Rules**: Establishing criteria for removing irrelevant content, such as template lines in Wikipedia, or determining when to exclude a document or line based on the proportion of symbols.
> - **Content Formatting**: Deciding how to structure and organize content, for example, how to handle the organization of questions and answers in StackExchange.
> - **Deduplication Parameters**: Choosing the thresholds and parameters for the MinHash LSH algorithm used for deduplication.
>
> Assessing the quality of the data produced under these decisions is an open and challenging problem. The most straightforward approach is to evaluate it through pre-training results. However, many decisions may not directly reflect improvements in pre-training outcomes, and running extensive pre-training experiments is resource-intensive. As a result, we relied heavily on insights from previous work (e.g., rules and threshold settings) and refined these rules and thresholds through manual inspection and adjustments after observing cleaned samples. This process ultimately led to the creation of MathPile.
>
> We hope this explanation clarifies the decisions involved in creating the pre-training corpus. We will provide a more comprehensive explanation in the next version of the paper.

---

> ### Author Rebuttal · Authors · 2024-08-17
>
> We sincerely appreciate the insightful comments from you! We respond to your comments below.
>
> > Also, the scripts are missing - it seems impossible to reproduce the dataset.  Why have you not made the cleaning scripts available?
>
> We completely understand your concern regarding the reproducibility of our dataset. **In fact, we had already open-sourced data-cleaning scripts more than two months ago.** You can find them at our GitHub repo of MathPile: https://github.com/GAIR-NLP/MathPile/tree/main/src  We apologize if these were overlooked. We  sincerely invite you to review the scripts, and we hope this clears up any confusion.
>
> > I am missing a section on which parts of the data were cleaned by scripts and which by humans (if at all).
>
> Thank you for your valuable suggestion. We have summarized the parts of the dataset collection (cf. `Section 2`) and global data preprocessing (cf. `Section 3`) where human intervention was involved and whether the cleaning process was automated in the table below. We hope this provides a clearer understanding of our dataset collection process. We will also include a dedicated section in the next version of the paper to address this aspect.
>
> | **MATHPILE Subset**                | **Human Involvement in Data Collection**                                                                     | **Cleaning Process Automated?**                                                          |
> |------------------------------------|--------------------------------------------------------------------------------------------------------------|------------------------------------------------------------------------------------------|
> | **Textbooks**                      | Manual search and download of open-source, free mathematics textbooks; quality check; setting cleaning rules | Yes, the automated application of the PDF conversion API and the document cleaning rules |
> | **arXiv Papers**                   | Manual selection of relevant mathematical field categories; setting latex cleaning and formatting rules      | Yes, automated cleaning steps like comment removal and format conversion                 |
> | **Wikipedia Mathematical Entries** | Humans observed samples to define rules for cleaning irrelevant content such as copyright statements         | Yes, automated HTML to Markdown conversion; and removal of extraneous lines              |
> | **ProofWiki Entries**              | No significant human intervention (mainly data dump selection, reformatting design)                           | Yes, automated text parsing and formatting                                               |
> | **StackExchange Discussions**      | Selection of relevant mathematics-related sites within the StackExchange network; setting filter thresholds  | Yes, automated HTML parsing and conversion, score filtering                              |
> | **Common Crawl Web Pages**         | Manual adjustment of TF-IDF rules to improve mathematical content identification                             | Yes, automated application  of rule-based mathematical document filtering                |
>
>
> | **Global Data Preprocessing Step** | **Human Intervention**                                             | **Cleaning Process Automated?**                                       |
> |------------------------------------|--------------------------------------------------------------------|-----------------------------------------------------------------------|
> | **Language Identification**        | Manual adjustment of thresholds based on observed false positives  | Yes, using FastText with custom thresholds                            |
> | **Data Cleaning & Filtering**      | Human observation to define rules for filtering irrelevant content | Yes, automated application of rules for content filtering and removal |
> | **Data Deduplication**             | Manual review of near-duplicate samples from different sources     | Yes, automated using MinHash LSH.                                     |
> | **Data Contamination Detection**   | Human verification of flagged benchmark duplicates                 | Yes,  automated detection based on exact-match hashing                |

---

> ### Author Response · Authors · 2024-08-31
> **Seeking Further Feedback**
>
> Dear Reviewer W3kS,
>
> I hope you're doing well. As the discussion period is nearing its end, I wanted to thank you once again for your valuable feedback. We believe that we have addressed your concerns through the supplementary details and clarifications regarding the construction of MathPile.
>
> If you still have any further reservations or suggestions, please don't hesitate to share them. Your insights are invaluable to us, and we're keen to address any remaining issues.
>
> Best regards,
>
> Authors

---

> ### Comment · Reviewer_W3kS · 2024-09-03
>
> Thank you for these clarifications. So in principle the dataset is actually reproducible; now the issue is to make any reproducible effort _useable_. In this regard:
>
> >  However, once these decisions were made, they were codified into our scripts, which are openly available and executed automatically.
>
> Please include much more comprehensive explanations about what each script does and in what ways decisions were codified into scripts. Appendix E is only about half a page (or, conversely, if this type of filtering was all that was carried out, please state so). Given the comprehensive nature of the dataset, I would expect Appendix E to be at least a page long and contain a paragraph describing exactly each script file, input, output, and the changes that it makes at a high level.
>
> While the evaluation is still below the standard, using a single LLM, I am happy to raise my score if such edits as explained in the previous paragraph are made.

---

> > ### Author Response · Authors · 2024-09-05
> > **Rebuttal by Authors (1/N)**
> >
> > Thank you once again for your thoughtful and constructive feedback. We greatly appreciate the time and effort you have invested in helping us refine and improve our work. We would like to offer the following clarifications and commitments to address your concerns.
> >
> > We fully agree that a more detailed description of each script, including its input, output, and the specific decisions codified within, is essential for enhancing the reproducibility and usability of our dataset.
> > To this end, here we provide a comprehensive, step-by-step explanation of each script across each stage of our MathPile construction, making it more transparent and reproducible. These will be added in Appendix E in our next version. The construction process of MathPile includes the corpora collection phase (refer to `Sec. 2`) in which some necessary data preprocessing and prefiltering processes are included, and the global data processing phase (refer to `Sec. 3`).
> >
> > ## The corpora collection phase
> >
> > The necessary details will be introduced step by step, according to each data source.
> >
> > ### (1) arXiv
> >
> > **Stage I: preparation for cleaning**
> >
> > - 1.1.1 classifying and processing different file types from a dataset, such as .tar.gz, .tex.gz, .pdf, and other .gz files, specifically from the arXiv dataset.
> >   - **Input**: Given an arXiv directory containing the raw dataset files from arXiv with various file types, such as compressed files and PDFs as input,
> >   - **Functional description**: this script firstly automatically detects the type of each file, moves the file into the appropriate pre-defined directory based on its extension, and then decompresses the compressed files, and saves them in the appropriate directory.
> >   - **Output**: The processed files are organized into different directories and each directory contains files that have been renamed, decompressed if necessary, and sorted according to their type. The script also provides a final summary showing how many files were processed for each file type.
> >
> > - 1.1.2 Copying and merging LaTeX (.tex) files from various directories into a target directory.
> >   - **Input**: The script processes files from multiple directories, which contain .tex and compressed files.
> >   - **Functional Description**: The script identifies LaTeX files (.tex) or directories containing such files and copies them to a target directory. It avoids copying duplicates and uses multi-threading to process files concurrently.
> >   - **Output**: .tex files and relevant directories are copied to the target directory, with duplicates skipped, and a log is produced once the copying is complete.
> >
> > **Stage II. Cleaning and processing LaTeX files from the arXiv dataset.**
> > - **Input**: The script processes .tex files from a raw arXiv dataset directory. These files may be stored as standalone .tex files or within directories that need to be merged and cleaned.
> > - **Functional Description**:
> >   - This script detects, cleans, and processes LaTeX files by removing comments, extracting titles, replacing custom commands, cleaning formatting, and handling figure, acknowledgements, and bibliography sections.
> >   - It automatically merges multiple .tex files from directories, replacing \input{} and \import{} commands with the actual content from the referenced files.
> >   - The script uses multi-threading to process files concurrently and includes timeout mechanisms to handle long-running processes.
> >   - Some critical decisions include:
> >     - Comment Removal and Custom Command Replacement: Uses regex to identify and replace comments and custom LaTeX commands to clean the document.
> >     - Timeout Threshold: Set to 5 minutes (300 seconds) to ensure timely completion of the cleaning process.
> > - **Output**: Cleaned .tex files are saved in a specified output directory, with the processed content stripped of unnecessary elements (e.g., comments, excess formatting) and merged properly. Logs track how many directories lacked .tex files or main document files.

---

> > > ### Author Response · Authors · 2024-09-05
> > > **Rebuttal by Authors (2/N)**
> > >
> > > ### (2) Textbooks
> > >
> > > **Stage I: Preparation for PDF Parsing**
> > >
> > > - **Input**: The script processes files from specified directories (different levels, such as K12 and College) and scans for .pdf files.
> > > - **Functional Description**: The script calculates the MD5 hash of each .pdf file to check for duplicates. If the MD5 hash of a file matches an existing entry in the dictionary, it identifies the file as a duplicate.  The script logs the duplicate files and counts the total number of duplicates identified.
> > > - **Output**: The script reports how many .pdf files are identical, logs the file paths of the duplicates and deletes duplicate files if uncommented.
> > >
> > > **Stage II: PDF Parsing and check download completions**
> > > - **Input**: The scripts take a directory of PDF files as input. Each file is sent to the Mathpix API for conversion, and the resulting data is downloaded in various formats (.md, .docx, .tex.zip, etc.).
> > > - **Functional Description**:
> > >   - The script uploads PDFs to the Mathpix API, tracks the processing status, and retrieves the results in multiple formats (Markdown, LaTeX, DOCX, etc.). It checks for the completeness of downloads and processes files within specified size limits.
> > >   - Key decisions include the use of file size thresholds (e.g., PDFs larger than 40MB are skipped) and waiting times (e.g., 10 seconds between status checks) to ensure efficient API interaction and file processing. For larger files, we use tools to compress and split these PDFS to fit the API call requirements
> > > - **Output**: The scripts save processed results in various formats to the specified directories, ensuring that each file has all necessary formats downloaded. Logs provide feedback on successful downloads and missing files.
> > >
> > > **Stage III: TextBooks cleaning generated from Mathpix-parsed PDF results**
> > >   - **Input**: the directories including Markdown (or LaTex) files
> > >   - **Functional Description**:
> > >     - The script uses regex-based patterns to clean sections like Preface, Table of Contents, Acknowledgements, and Index, as well as remove Mathpix image URLs. It also replaces excessive newlines to improve formatting.
> > >     - Multi-threading is used to clean multiple files concurrently, and file encoding is detected to handle various text formats.
> > >     - Key Decisions include regex patterns for identifying and removing specific sections, and handling content structure (e.g., replacing Mathpix image links and removing unwanted sections).
> > >   - **Output**: The cleaned files are saved in a target directory, ensuring that unnecessary sections and content are removed. The script also logs processing time and actions taken for each file.
> > >
> > >
> > > ### (3) StackExchange
> > >
> > > **Stage I: Processing XML files from Stack Exchange sites and converting them to plain text.**
> > > - **Input**: The script processes XML files (e.g., Posts.xml) from various Stack Exchange sites like `math.stackexchange.com`, and others. These files contain posts, including both questions and answers.
> > > - **Functional Description**:
> > >   - The script parses the XML files to extract posts and filters only questions and answers (PostTypeId 1 and 2). It cleans the HTML content from the body of each post using BeautifulSoup, removes invalid XML characters, and converts the posts into plain text. The cleaned posts are written into a JSONL file, with one JSON object per line.
> > > - **Output**: The cleaned and filtered posts are saved as JSONL files in the target directory, with the filename corresponding to the original Stack Exchange site. The script logs the successful writing of files and provides the count of processed posts for each site.
> > >
> > > **Stage II: Pairing questions and answers from preprocessed Stack Exchange posts**
> > > - **Input**: The script reads JSONL files from a directory, which contain preprocessed posts (questions and answers) from various Stack Exchange sites.
> > > - **Functional Description**:
> > >   - The script first reads the posts and filters out unnecessary metadata fields. It then pairs questions (PostTypeId 1) with their corresponding answers (PostTypeId 2), organizing them into a dictionary where each question is mapped to its accepted answer (if available) and other answers.
> > >   - Key decisions include handling the logic for accepted answers and ensuring no duplicate answers are associated with a question.
> > > - **Output**: The paired question-answer sets are saved as JSONL files in a target directory, with one set per line. The script logs the number of paired question-answer sets for each site and confirms the successful writing of the output files.

---

> > > > ### Author Response · Authors · 2024-09-05
> > > > **Rebuttal By Authors (3/N)**
> > > >
> > > > **Stage III: Filtering and processing Stack Exchange posts based on question and answer scores**
> > > >
> > > > - **Input**: The script reads preprocessed JSONL files containing Stack Exchange posts (questions and answers) from directories obtained in Stage II.
> > > > - **Functional Description**:
> > > >   - The script checks and cleans each post by removing unnecessary metadata (e.g., CommentCount, ContentLicense). It filters questions and answers based on their scores with custom thresholds. Only questions and answers with scores above these thresholds are retained.
> > > >   - Key Decisions: After human check, the thresholds for questions and answers are both set as 5. Questions with a score below 5 are filtered out, and answers must meet a minimum score, which may vary depending on the accepted answer score.
> > > > - **Output**: The filtered posts are written into JSONL files in the target directory, with logs showing the number of questions and answers before and after filtering.
> > > >
> > > >
> > > > ### (4) Wikipedia
> > > > Processing Wikipedia ZIM archives and converting HTML pages to cleaned Markdown format.
> > > >
> > > > - **Input**: The script processes .zim files (e.g., wikipedia_en_mathematics_nopic_2023-08.zim), which contain compressed HTML data from Wikipedia pages.
> > > > - **Functional Description**: The script reads the .zim archive, identifies HTML pages and decodes their content. It processes LaTeX math elements within the HTML and converts the content to Markdown using text extraction tools (html2text or Resiliparse), removing unwanted elements such as images, excessive newlines, and Wikipedia footer templates.
> > > >   - Some Key Decisions:
> > > >     - LaTeX Processing: Converts `<math>` HTML tags to LaTeX syntax.
> > > >     - Markdown Cleanup: Removes images and footers, consolidates newlines, and ensures the text is formatted correctly for Markdown.
> > > > - **Output**: The cleaned Markdown content for each Wikipedia page is saved as a JSONL file, where each entry includes metadata (e.g., title, path) and the cleaned Markdown content. The output is logged to confirm the successful generation of each cleaned document.
> > > >
> > > >
> > > > ### (5) Common Crawl
> > > >
> > > > **Stage I: Gather common crawl data from SlimPajama-627B**
> > > >
> > > > Filtering and chunking large JSONL datasets from multiple directories.
> > > > - **Input**: The script processes .jsonl files from a specified directory, which contains the whole dataset of SlimPajama-627B.  Each line in the .jsonl file is an individual data point with metadata.
> > > > - **Functional Description**: The script traverses the directory to find all .jsonl files and loads their content. It only retains the RedPajamaCommonCrawl and RedPajamaC4 subsets. Once a chunk reaches the size of 1,000,000 entries, the script writes the chunk to a new .jsonl file. The process repeats until all files are processed.
> > > > - **Output**: Filtered data chunks are written to .jsonl files, with logs indicating the number of chunks written.
> > > >
> > > > **Stage II: Filtering and chunking mathematical content from large JSONL datasets.**
> > > > - **Input**: The script processes .jsonl files from a directory obtained in stage I, which contains large text datasets.
> > > > - **Functional Description**: The script traverses the directory, loading .jsonl files and filtering text entries that match specific mathematical content patterns. It identifies text entries that contain mathematical expressions using predefined patterns (e.g., `\math`, `\frac{`, etc.). It uses multi-processing (Pool) to process chunks of files concurrently, taking advantage of multiple CPU cores to speed up the filtering process.
> > > >   - Key Decisions include leveraging predefined patterns (e.g., `\math`, `\frac{`, etc.) to identify potential math documents.
> > > > - **Output**: Filtered text entries with mathematical content are written in .jsonl files in chunks of 1,000,000 entries each with logs indicating successful writing.

---

> > > > > ### Author Response · Authors · 2024-09-05
> > > > > **Rebuttal By Authors (4/N)**
> > > > >
> > > > > ### (6) ProofWiki
> > > > >
> > > > > Parsing, cleaning, and structuring ProofWiki data for theorem retrieval dataset generation.
> > > > > - **Input**: The script processes a large XML dump from ProofWiki, identifying and parsing pages related to theorems, definitions, proofs, and other related content.
> > > > > - **Functional Description**: The script first identifies redirects and links them correctly to avoid broken links. It processes various page types, including theorems, definitions, and proofs, by identifying specific sections (e.g., "== Theorem ==", "== Proof ==") and extracting relevant content like links, categories, and the text itself. Theorem pages are linked with their corresponding proofs and references. After parsing, the script filters out pages without relevant content (e.g., unfinished proofs marked as {{ProofWanted}}) and ensures that parsed content is correctly linked to theorems and definitions. The script performs a deduplication step to remove identical or highly similar theorems and proofs by comparing their content. Each reference (theorem, definition, proof) is assigned a unique identifier, which is included in the final dataset along with the links between theorems and their proofs.
> > > > > - **Output**: The structured data is saved in a JSON format, including categorized content such as theorems, definitions, proofs, and retrieval examples, each linked to its corresponding references and proofs. The dataset includes metadata like unique identifiers, categories, and the cleaned content of theorems, definitions, and proofs.
> > > > >
> > > > >
> > > > > ## Global Data Processing
> > > > >
> > > > > After the collection phase, we will perform global data processing to improve data quality further.
> > > > >
> > > > > ### (1) Language Identification
> > > > >
> > > > > - **Input**: The script processes .jsonl files containing datasets from different sources and applies FastText's language identification model (lid.176.bin) to filter English content.
> > > > > - **Functional Description**: The script first loads .jsonl files, splits them into chunks of a specified size (default is 5000), and creates directories for saving processed output. Using FastText, the script evaluates the language of each document or question-answer pair (for StackExchange data). Only documents with an English language detection score higher than a threshold are retained. The script parallelizes language detection, running multiple processes concurrently to speed up processing. During processing, the script tracks and logs the number of English documents, total documents processed, and average language detection score across all documents.
> > > > >   - Key Decisions: we set customized language thresholds for each data source after manual observation of some filtering results. Specifically, Wikipedia and StackExchange thresholds were set at 0.1, arXiv at 0.3, and Common Crawl at 0.5. No thresholds were applied to ProofWiki and Textbooks due to manual verification ensuring English content.
> > > > > - **Output**: Filtered documents with English content are written in a new .jsonl file in the target directory, and logs provide insights into the number of English documents and processing times for each dataset.

---

> > > > > > ### Author Response · Authors · 2024-09-05
> > > > > > **Rebuttal By Authors (5/N)**
> > > > > >
> > > > > > ### (2) Data Cleaning and Filtering
> > > > > > - **Input**: The script processes .jsonl files from a specified directory, which contain datasets that have already been filtered for English language content using FastText's language identification model.
> > > > > > - **Functional Description**: The script aims to filter out low-quality or irrelevant documents from large datasets by applying a series of heuristic rules. These rules are inspired by standards from large-scale datasets like Gopher and RefinedWeb, specifically crafted for the mathematical domain. The goal is to ensure that the resulting data is of high quality, containing meaningful and coherent text suitable for training language models.
> > > > > > - **Filtering Rules and Key Decisions**
> > > > > >   - **Rule 1: Detecting and Removing "Lorem Ipsum" Placeholder Text**
> > > > > >     - The script checks if the normalized content (converted to lowercase) contains the phrase "lorem ipsum." If found, it examines each line and removes lines that consist primarily of "lorem ipsum" with few additional characters.
> > > > > >     - Key Decision: Lines that are mostly placeholder text are discarded to improve the overall quality of the document.
> > > > > >   - **Rule 2: Filtering JavaScript-Related Lines**
> > > > > >     - This rule is aimed at removing lines that are browser messages or prompts related to JavaScript, which are irrelevant to the content. Specifically, the script detects lines containing the word "javascript" along with keywords like "enable," "disable," or "browser," especially if the line length is less than 200 characters. Such lines are removed from the document.
> > > > > >     - Key Decision: Filtering out these lines prevents inclusion of non-content messages in the dataset. The setting of auxiliary keywords like "enable" is crucial.
> > > > > >   - **Rule 3: Removing Short Lines with Specific Patterns**
> > > > > >     - This rule targets eliminating lines that are likely to be navigation prompts or incomplete content. Specifically, lines shorter than 10 words containing phrases like "Login," "sign-in," "read more...," or "items in cart" are identified and removed.
> > > > > >     - Key Decision: These lines are often irrelevant and do not contribute meaningful information. The consideration of keywords like "Login" and line length are also crucial.
> > > > > >   - **Rule 4: Uppercase Word Fraction**
> > > > > >     - This rule identifies documents with an unusually high proportion of uppercase words, which may indicate poor formatting or spam. Specifically, the script calculates the fraction of words in the document that are entirely uppercase. If this fraction exceeds 40%, the document is discarded.
> > > > > >     - Key Decision: A high uppercase ratio is a strong signal of low-quality content. The uppercase ratio threshold is set after manual observation.
> > > > > >   - **Rule 5: Ellipsis Fraction**
> > > > > >     - This rule detects documents overusing ellipses (...), which may indicate incomplete or informal content. Specifically, it calculates the fraction of lines ending with an ellipsis. Documents where this fraction exceeds 30% are discarded.
> > > > > >     - Key Decision: Excessive use of ellipses suggests that the content may be unsuitable for training. The  fraction threshold of lines ending with an ellipsis is set after manual observation.
> > > > > >   - **Rule 6: Non-Alphabetical Characters Fraction**
> > > > > >     - This rule filters out documents dominated by symbols or numbers rather than alphabetic characters. Specifically, the script computes the fraction of words without any alphabetic characters. Documents with a non-alphabetic word fraction exceeding 80% are removed.
> > > > > >     - Key Decision: Ensuring that documents contain primarily natural language text. The non-alphabetic word fraction is set after manual observation.
> > > > > >   - **Rule 7: Mean Length of Alphabetic Words**
> > > > > >     - This rule ensures that the document's words are of reasonable length, as extremely short or long words may indicate anomalies. Specifically, calculates the average length of words consisting only of alphabetic characters. Documents with an average word length of less than 3 or greater than 10 are discarded.
> > > > > >     - Key Decision: Filters out documents that may contain gibberish content. The average length range of words is set after manual observation.
> > > > > >   - **Rule 8: Presence of Common Stop Words**
> > > > > >     - This rule verifies that the text resembles natural language by checking for common English stop words. Specifically, it checks if the content contains at least two of the following stop words: "the," "be," "to," "of," "and," "that," "have," "with." Documents lacking at least two stop words are discarded.
> > > > > >     - Key Decision: Text without common stop words is unlikely to be natural English prose.

---

> > > > > > > ### Comment · Reviewer_W3kS · 2024-09-05
> > > > > > >
> > > > > > > That looks good to me, I have updated my score.

---

> > > > > > > > ### Author Response · Authors · 2024-09-05
> > > > > > > >
> > > > > > > > Thanks very much!

---

> > > > > > > ### Author Response · Authors · 2024-09-05
> > > > > > > **Rebuttal By Authors (6/N)**
> > > > > > >
> > > > > > > - **Rule 9: Symbols-to-Words Ratio**
> > > > > > >     - This rule identifies documents with a high proportion of symbols, which may indicate code snippets or malformed text. Specifically, it calculates the ratio of certain symbols (e.g., ..., …) to the total number of words. Documents with a symbols-to-words ratio exceeding 50% are discarded.
> > > > > > >     - Key Decision: High symbol content may reduce the quality and readability of the text. The symbols-to-words ratio threshold is set after manual observation.
> > > > > > >   - **Rule 10: Lines Starting with Bullet Points**
> > > > > > >     - This rule is designed to avoid documents that are predominantly bullet-point lists rather than continuous prose. Specifically, it determines the fraction of lines that start with common bullet characters (e.g., •, –). Documents where more than 90% of lines start with bullet points are discarded.
> > > > > > >     - Key Decision: Ensures the dataset contains narrative text rather than lists. The fraction threshold is set after manual observation.
> > > > > > >   - **Rule 11: Minimum Character Count**
> > > > > > >     - This rule is aimed at ensuring documents have sufficient length to be meaningful. Specifically, it counts the number of characters after removing spaces, punctuation, and newline characters. Documents with fewer than 200 such characters are discarded.
> > > > > > >     - Key Decision: Short documents may not provide enough context or information. The threshold of characters is set after manual observation.
> > > > > > > - **Output**: The script writes the filtered documents to .jsonl files in the specified directory, preserving the same directory structure and filenames as the input. Each output file contains only the documents that pass all the filtering criteria, along with updated metadata fields indicating the results of various filtering steps (e.g., calculated ratios). For StackExchange data, only items where both the question and at least one answer pass the filtering are retained.
> > > > > > >
> > > > > > >
> > > > > > > ### (3) Deduplication
> > > > > > >
> > > > > > > **Stage I: Data Deduplication in each source**
> > > > > > >
> > > > > > > - **Input**: a directory that contains all data of a data source.
> > > > > > > - **Functional Description**: This script utilizes the MinHash LSH algorithm to remove repetitive documents in each source, which is vital for training efficiency and reducing data memorization. Specifically, the script involved splitting each document using whitespace and constructing 5 grams, applying the “sha1” hash function, and configuring 450 buckets with 20 minhashes each, totalling 9,000 minhashes per document, as per RefinedWeb’s guidelines.
> > > > > > >   - Key decisions: The parameters of MinHash LSH are set after manual observation.
> > > > > > > - **Output**: The script writes the deduplicated documents to .jsonl files in the specified directory, preserving the same directory structure and filenames as the input.
> > > > > > >
> > > > > > > **Stage II: Data Deduplication across sources**
> > > > > > > - **Input**: all data that have been deduplicated internally from all data sources.
> > > > > > > - **Functional Description**: This script continues to employ the MinHash LSH algorithm with the same parameters to remove repetitive documents.
> > > > > > > - **Output**: The script writes the globally deduplicated documents to .jsonl files in the specified directory, preserving the same directory structure and filenames as the input.
> > > > > > >
> > > > > > >
> > > > > > > ### (4) Data Contamination Detection and Removal
> > > > > > > - **Input**: The corpus that has undergone the above processes and all target math benchmark test sets.
> > > > > > > - **Functional Description**: This script performs line-level deduplication of text data using exact hashing. The goal is to remove duplicate lines with benchmark test sets from our corpus at a very granular level, where each line in a document is hashed and checked for duplicates across the dataset. The script uses various hash functions and processes large datasets efficiently with parallel processing.
> > > > > > >   - **Key decision**: Before hashing, each line is normalized to standardize the content (e.g., removing extra spaces, lowercasing text). This increases the chances of detecting duplicates that might otherwise differ due to minor variations in formatting.
> > > > > > > - **Output**: The script writes the decontaminated corpus into JSONL files in the specified directory, preserving the same directory structure and filenames as the input.

---

> > > > > > > > ### Author Response · Authors · 2024-09-05
> > > > > > > > **Rebuttal By Authors (7/N)**
> > > > > > > >
> > > > > > > > We've made every effort to thoroughly explain all the essential details involved in building MathPile, including the input and output of each script, functional descriptions, and key decisions made along the way. We hope this clarifies any questions you may have.
> > > > > > > >
> > > > > > > > **About the experiments**, we fully understand the importance of comprehensive evaluations across multiple models. However, due to limited computational resources, we were only able to conduct experiments using Mistral, which was a state-of-the-art open-source 7B model at the time, to validate the effectiveness of our approach. We are actively seeking collaborations to expand our experiments in the future. We hope you can appreciate our efforts in producing, cleaning, and curating a high-quality dataset, as well as its necessity for advancing this field.
> > > > > > > >
> > > > > > > > Finally, we truly appreciate the effort and time you have taken to review our work and hope these address your concerns.

---

### Comment · Area_Chair_UPJM · 2024-08-29
**Please engage in the author-reviewer discussion**

Dear Reviewers,

Thank you for your hard work on the papers and reviews. Please note that the deadline for the author-reviewer discussion period is approaching (August 31, 2024). Some of you have engaged in discussions with authors-thank you! For reviewers who have not yet, please discuss with authors as this is a very important part of the reviewing process and authors are eager to have further feedback from you. If there are any changes to your scores, kindly provide explanations for these adjustments.

---

### Decision · Program_Chairs · 2024-09-26

**Decision:**

Accept (Poster)

**Comment:**

The paper presents MathPile, a large-scale (9.5B tokens) high-quality and diverse corpus for model pre-training with a focus on mathematical reasoning.

Pros:
1. The paper is well motivated and presented.
2. The curated dataset is of high quality and diverse, using a rigorous data processing pipeline.
3. The paper includes a robust data contamination detection mechanism to eliminate duplicates.

Cons:
1. It would be better to experiment with more base models.
2. A clear and verified data mixing mechanism would make the dataset more usable.